# Implicit neural representations for accurate estimation of the Standard Model of white matter

Tom Hendriks [1,4] ✉, Gerrit Arends [2,4], Edwin Versteeg[2], Anna Vilanova[1], Maxime Chamberland[1,5] & Chantal M. W. Tax[2,3,5]

Diffusion magnetic resonance imaging (dMRI) enables non-invasive investigation of tissue microstructure. The Standard Model (SM) of white matter aims to disentangle dMRI signal contributions from intra- and extra-axonal water compartments. However, due to the model's high-dimensional nature, accurately estimating its parameters poses a complex problem and remains an active field of research, in which different (machine learning) strategies have been proposed. This work introduces an estimation framework based on implicit neural representations (INRs), which incorporate spatial regularization through the sinusoidal encoding of the input coordinates. The INR method is evaluated on both synthetic and in vivo datasets and compared to existing methods. Results demonstrate superior accuracy of the INR method in estimating SM parameters, particularly in low signal-to-noise conditions. Additionally, spatial upsampling of the INR can represent the underlying dataset anatomically plausibly in a continuous way. The INR is self-supervised, eliminating the need for labeled training data. It achieves fast inference, is robust to noise, supports joint estimation of SM kernel parameters and the fiber orientation distribution function with spherical harmonics orders up to at least 8, and accommodates gradient non-uniformity corrections. The combination of these properties positions INRs as a potentially important tool for analyzing and interpreting diffusion MRI data.

Diffusion magnetic resonance imaging (dMRI) is a non-invasive technique for in vivo measurement of water diffusion in tissue using magnetic field gradients. To extract biologically interpretable information, a common approach is to fit a microstructural tissue model to a set of signals acquired with different dMRI acquisition settings[1–4]. In the absence of diffusion time dependence, these typically include different combinations of gradient strengths (commonly quantified by the b-value), directions (b-vector), and B-tensor shape[5]. Microstructural parameters estimated by these models – including compartmental signal fractions and diffusivities – have shown to be sensitive to changes in brain structure due to diseases like multiple sclerosis[6], Alzheimer's disease[7] and Parkinson's disease[8], and can provide a more fundamental understanding of tissue microstructure in both healthy and pathological tissues[9].

The Standard Model of white matter (SM), see ref. 4 for a review, describes the signal arising from white matter by a kernel consisting of three compartments (intra-axonal, extra-axonal, and free water (occasionally omitted)) convolved with a fiber orientation distribution (FOD)[10]. Compartmental signal fractions and diffusivities can be estimated, alongside the parameters that describe the FOD (usually in the form of a spherical harmonics (SH) series). Nevertheless, the high-dimensional parameter space of the SM complicates the estimation of its parameters, potentially leading to low accuracy, precision, and degeneracy of estimates[11]. These issues become even more prominent at high noise levels.

Multiple strategies have been employed to fit microstructure models to dMRI data. When the primary goal is to estimate the tissue's directional structure, a common two-step approach involves first fixing the kernel using a global estimate, followed by solving a linear inverse problem to estimate the fiber orientation distribution (FOD)[12,13]. In contrast, when the focus is on estimating the kernel parameters, the orientational dependence is factored out by using

[1]Department of Computer Science and Mathematics, Eindhoven University of Technology, Eindhoven, The Netherlands. [2]Center for Image Sciences, University Medical Center Utrecht, Utrecht, The Netherlands. [3]Cardiff University Brain Research Imaging Centre (CUBRIC), School of Physics and Astronomy, Cardiff University, Cardiff, UK. [4]These authors contributed equally: Tom Hendriks, Gerrit Arends.[5]These authors jointly supervised this work: Maxime Chamberland and Chantal M.W. Tax. ✉e-mail: t.hendriks@tue.nl

rotational invariants of the signal[14–17]. This last approach is most common for SM parameter estimation.

Estimation of SM parameters has been improved by machine-learning based methods including Bayesian estimators[16], neural networks[18,19], and fitting cubic polynomials[15,20]. Importantly, these approaches are commonly supervised machine-learning methods, operating at a voxel-level, that are fit by using simulated ground truth parameters and their associated signals – the training dataset. While this can be very effective, the quality of the results depend on biases existing in the training and inference datasets[21]. Additionally, since the methods operate at a voxel-level, they do not make any use of the spatial correlation that is naturally present in anatomy.

Recently, implicit neural representations (INRs) have been introduced to the dMRI domain as a novel self-supervised fitting method, which – rather than on a voxel level – fit models on a continuous space of coordinates and are trained on the dMRI signal directly without the realization of ground truth parameters. INRs have shown to create noise-robust continuous representations of dMRI datasets of individual subjects, using the spatial correlations present in the data. So far, they have been used to represent the diffusion signal using SH basis functions and to estimate the parameters of (multi-shell multi-tissue) constrained spherical deconvolution (MSMT-CSD)[22–24]. In this previous work, INRs demonstrate potential to improve on voxel-based methods to estimate parameters, especially in more noisy acquisitions. Since INRs are spatially regularized continuous representations of the dataset, they can potentially be beneficial when performing downstream tasks which require interpolation, such as microstructure-informed fiber tracking[25,26].

Building upon our previous work[22,24], we implement INRs to estimate the SM parameters alongside the FODs, and demonstrate the noise-robustness, continuous representation, and applicability of INRs for fitting on both synthetically generated and in vivo dMRI data. Synthetically generated datasets facilitate a quantitative analysis of the model outputs, while in vivo data quantitatively shows the performance in a realistic setting. The INR is compared to two existing machine learning methods for fitting the SM (Standard Model Imaging Toolbox (SMI)[15] and a supervised neural network (NN)[18], as well as nonlinear least squares (NLLS). Additionally, moving beyond existing methods, the FOD SH-coefficients up to order eight are estimated directly alongside the SM kernel parameters (intra-axonal diffusivity ($D_i$), extra-axonal axial diffusivity ($D_e^{\parallel}$), extra-axonal perpendicular diffusivity ($D_e^{\perp}$) and intra-axonal fraction ($f_i$)). Thus, in contrast to earlier methods that focused either on the kernel or on the FOD, this approach performs a joint estimation, which can improve accuracy and ameliorate degeneracy[27]. Moreover, every INR is fit on and represents a dMRI dataset of a single subject and, therefore, does not rely on a large number of training datasets as supervised methods do. Furthermore, it is capable of explicitly correcting for gradient non-uniformities by inputting the effective acquisition protocol (B-tensor) for each coordinate with the spatially varying gradient coil tensor of the scanner[28]. The latter would become impractical for supervised methods, which would need prohibitively large sets of training data to capture voxel-wise protocol deviations. Altogether, the proposed method provides a flexible, noise-robust, spatially coherent way of fitting the SM, which is self-supervised and, therefore, not biased by training data.

## Results

### Quantitative comparison on simulated data

Results from experiment 1 are presented in Fig. 1 (signal-to-noise ratio (SNR) = 20). When Gaussian noise is added, the INR method shows superior performance for all SM parameters, with both Pearson's correlation coefficient ($\rho$) and root mean squared error (RMSE) achieving the highest values. Brain parameter maps from Fig. 1b shows the ability of the INR method to reproduce smooth parameter maps similar to the ground truth, where the voxel-wise fitting methods show noisy estimates due to the lack of spatial regularization. The INR method does exhibit minor over-estimation of $D_e^{\parallel}$ in the splenium of the corpus callosum compared to the ground truth.

Parameter estimation on simulated data without noise and with SNR = 50 can be found in the supplementary information section 2 (Figs. S2 and S3), showing a less prominent, but still evident improvement in $\rho$ and RMSE of the INR method compared to other estimation methods for all parameters. Fitting without noise shows that the ground truth generating process is not positively biased towards parameter estimates of the INR.

### Rician noise bias

The correlation plots between the INR with mean squared error (MSE) or Rician loss and the ground truth parameters, alongside the parameter maps for both approaches, can be seen in Fig. 2. For MSE, the Rician bias is visible in the scatter plots and parameter maps as a general over- or under-estimation across all voxels. The Rician loss is able to correct the bias, resulting in better correlation with the ground truth. The bias is most significant for $D_i$ and $D_e^{\parallel}$. The bias effect is less prominent when considering SNR = 50, see supplementary information section 3, Fig. S4.

### Qualitative comparison on in vivo data

The parameter maps for the in vivo dataset for the different methods are seen in Fig. 3. The differences between methods are especially visible in the capsula interna and externa, and the splenium of the corpus callosum. The INR produces maps that are more spatially smooth and show a clear (and anatomically plausible) structure, while other methods display higher spatial variability.

### Estimation of SH order up to $l_{max}$ = 8

The INR outputs for the different SH order ($l_{max}$) FODs of the synthetic datasets are visualized in Fig. 4. Qualitative inspection of the FODs shows plausible FOD shapes and directions throughout all datasets and SH orders. Furthermore, there are no notable differences between the FOD estimates for the noiseless and SNR 50 synthetic dataset, indicating noise-robustness in the estimate. Small spurious peaks appear in the SNR 20 synthetic dataset, but the fiber orientations indicated by the larger peaks remains almost identical to both comparisons. In the in vivo dataset the INR produces plausible FOD shapes and directions as well, as visualized in Fig. 5. The backgrounds in Figs. 4 and 5 show no large voxel-wise differences for $f_i$ across the different orders. This holds true for all kernel parameter estimates, which is shown in further detail in the supplementary information section 4, Tables S1 and S2.

### Effect of gradient non-uniformity correction of SM parameter estimation

Brain parameter maps with and without gradient non-uniformity correction on in vivo data are presented in Fig. 6. Difference maps show a significant effect on $D_i$ and $D_e^{\parallel}$. Corrected maps show the effect at the edges of the brain, mainly in the frontal lobe. This is to be expected as gradient non-uniformity is strongest there. $D_i$ and $D_e^{\parallel}$ show lower values after correction. The influence of the correction is least apparent on $f_i$ and the rotational invariant of $l_{max}$ = 2 ($p_2$). The effect of the gradient non-uniformity correction is similar for both INR and NLLS fitting. Lower diffusivity values at the front and back of the brain appear using both methods, as well as higher $p_2$ values. The parameter $f$ shows only small differences between the approaches: NLLS shows no effect, while INR shows small corrections throughout the brain. Results of combining Rician bias loss with gradient non-uniformity correction can be found in the supplementary information section 5, Fig. S5.

### Implicit neural representation for spatial interpolation

The visualizations in Fig. 7 show the comparison between the $p_2$ parameter maps using different methods for upsampling. The linear interpolation maintains the pixelated appearance of the low-resolution data, especially visible in structures that are not aligned with the image grid (a 'staircase-like' effect). These artifacts are, although less prominent, still visible in cubic interpolation. The INR does not show these artifacts at this resolution, as the underlying continuous representation is less limited by the input resolution.

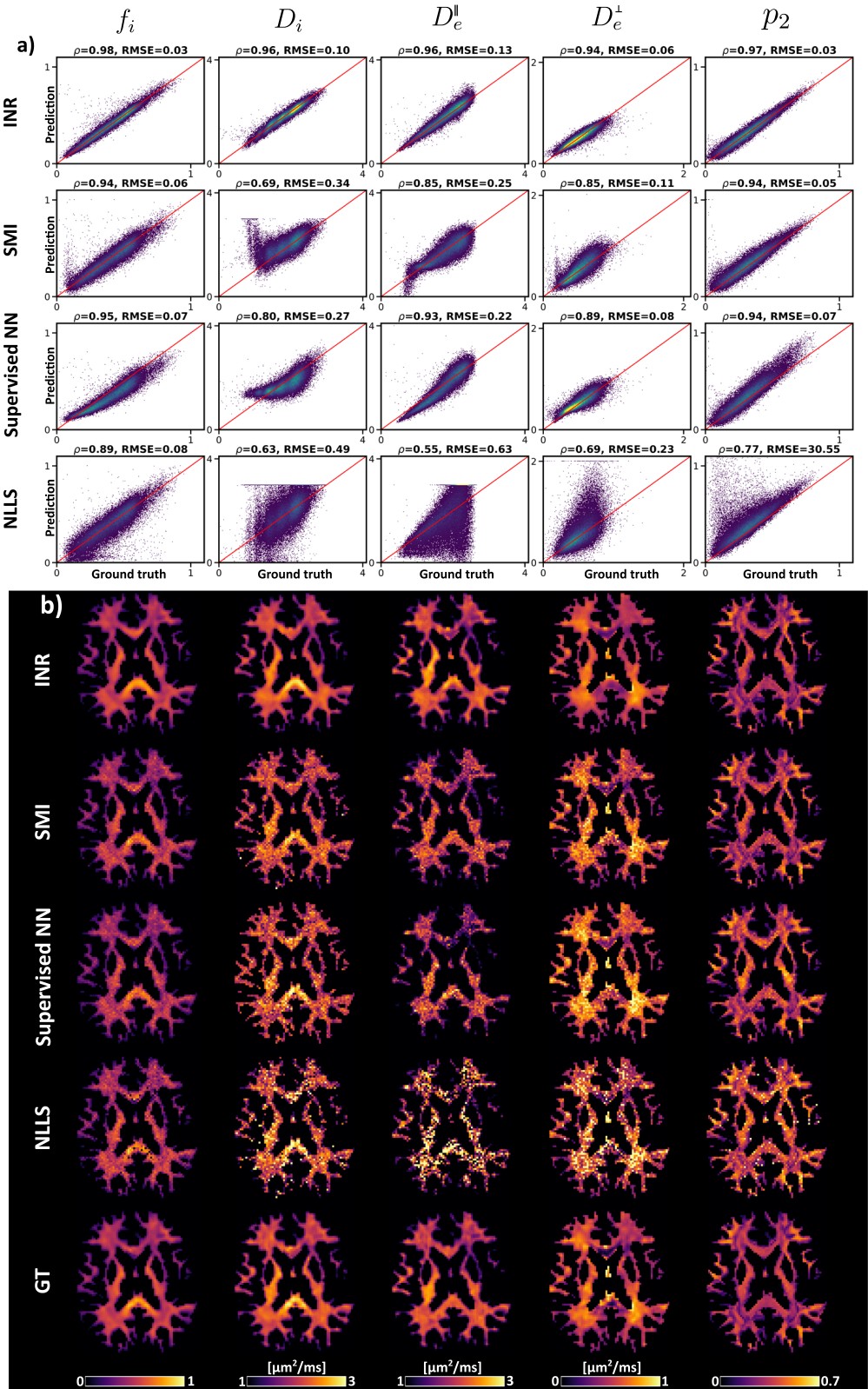

**Fig. 1 | Results of Standard Model fitting in experiment 1 (SNR 20). a** Scatter density plots of ground truth versus parameter estimations of all methods. The titles of the subplots indicate Pearson's correlation coefficient ($\rho$) and root mean squared error (RMSE). Every column corresponds to a specific parameter, which is indicated above. **b** SM parameter maps corresponding to the results in **a**. Bottom row shows the ground truth (GT).

## Model fitting times

Model fitting times for all INR experiments are shown in Table 1. The main influence on the fitting time is the size of the dataset (number of voxels in the WM mask), the size of the hidden layers, amount of epochs, the number of outputs (determined by $l_{max}$), and the usage of the analytical or numerical integration solution. The analytical approach was used in the experiments with the simulated dataset and the numerical approach for the in vivo data. The number of white matter voxels included in the

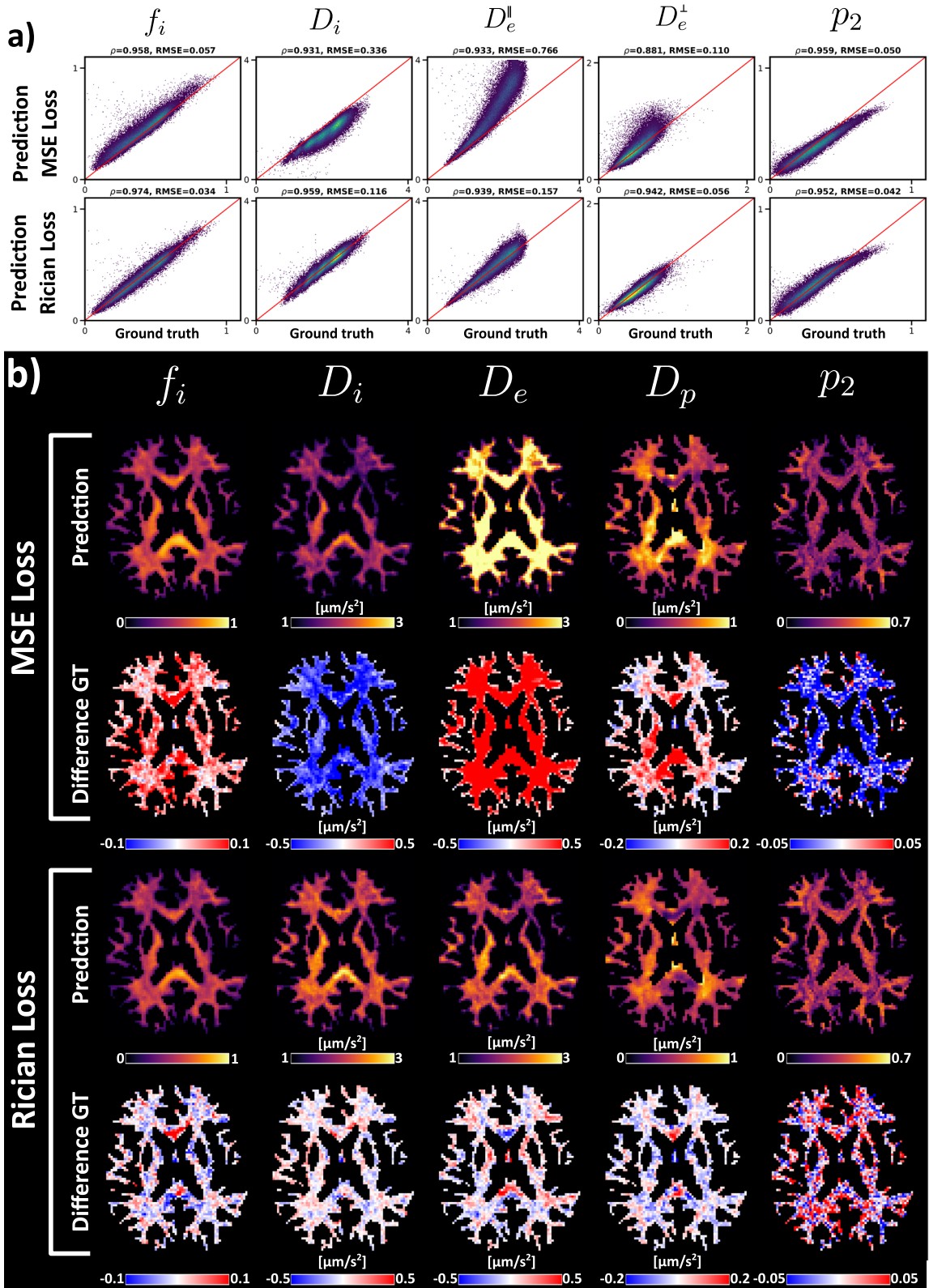

**Fig. 2 | Effect of Rician Loss Likelihood function on Standard Model fitting (SNR = 20). a** Scatter plots for estimation with MSE and Rician loss. Pearson correlation coefficient ($\rho$) and Root-Mean Squared Error (RMSE) are indicated in the subplot's title. Every column corresponds to a specific parameter, which is indicated above. **b** Brain parameter maps of the predictions from **a**. Difference maps are calculated with the ground truth (GT) parameters.

simulated data and in vivo data are 60800 and 11266, respectively. This means that the analytical approach is considerably faster than the numerical approach. The addition of gradient non-uniformity correction also increases fitting time.

## Discussion
### Implications of the results
In this work, we show how INRs can be used to estimate continuous, noise-robust SM parameter maps of simulated and in vivo datasets and

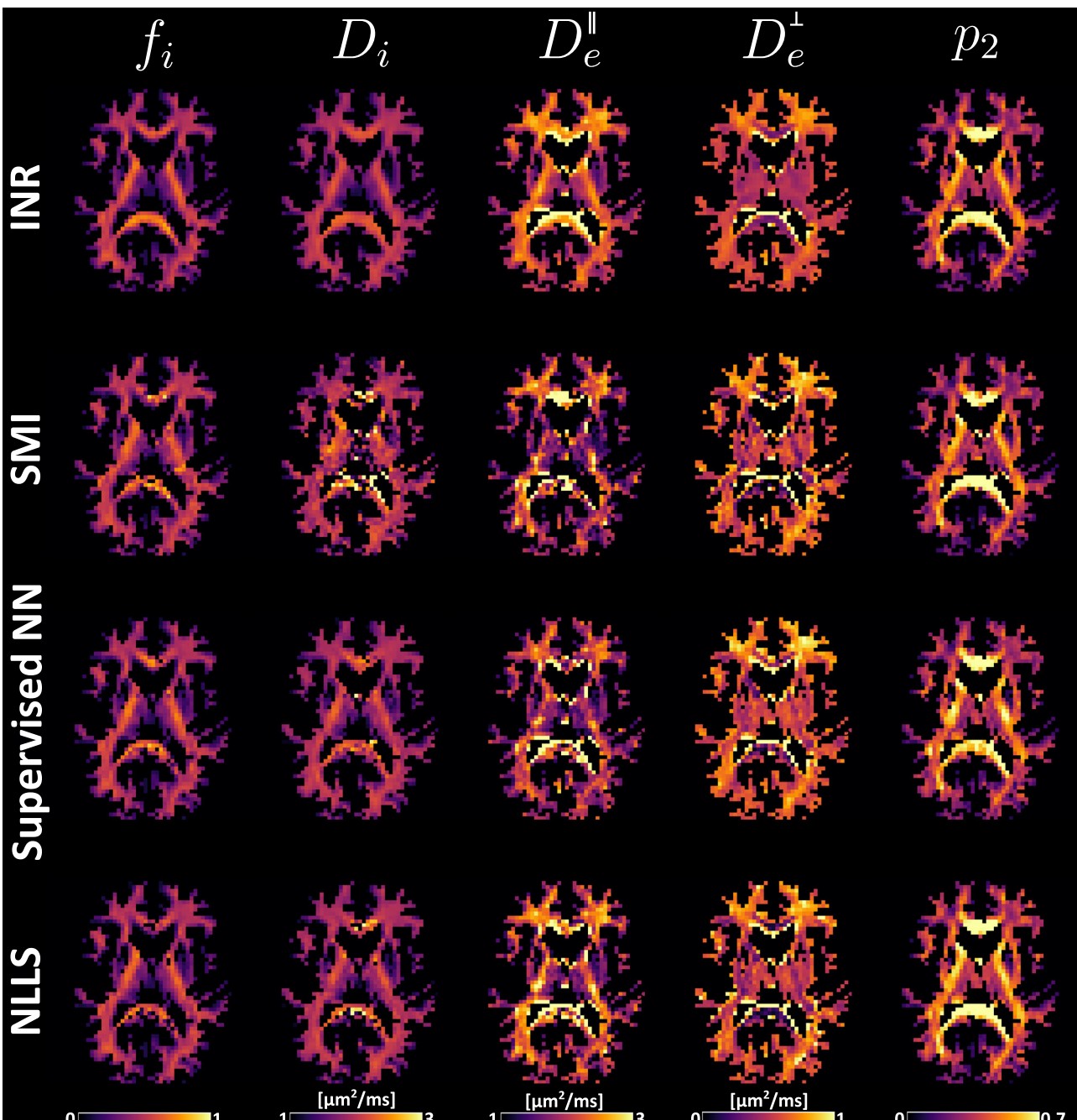

**Fig. 3 | Standard Model fitting results for in-vivo data for INR, SMI, Supervised NN and NLLS.** All SM parameters are plotted as single row. Every row corresponds to the fitting method indicated at the beginning of the specific row. All maps have equal scaling.

with FODs of different SH orders. The self-supervised, subject-wise nature of the framework prevents training set bias, while the continuous representation allows spatial correlations to improve parameter estimates and reduce the impact of noise. For high SNR levels, the supervised NN method achieves performance metrics close to those of the INR approach; however, it can be significantly more time-consuming due to its reliance on NLLS for generating part of the training data and the need to retrain the model for each specific acquisition protocol (see Supplementary Fig. S3). At higher noise levels (SNR = 20), the INR method clearly outperforms all other methods (see Fig. 1). On in vivo data, the underlying representation shows a more structurally correlated appearance, without large inter-voxel variability. This is further illustrated by upsampling the INR at high resolution. Parameter estimates

for FODs up to SH orders of at least eight can be provided alongside the other SM parameters without introducing bias in other parameters as shown in the supplementary information section 4. The self-supervised nature of the method avoids training set bias prevalent in supervised fitting methods.

The proposed hyperparameters $n_p = 5000$ and $n_h = 2048$ provide a robust setting that can provide good representations of dMRI datasets with different sizes, acquisition protocols, and levels of noise. An exploration of these hyperparameters can be found in the supplement of [24]. The hyperparameter $\sigma^2$ provides a convenient way of tuning the model to provide stronger or weaker spatial regularization, as detailed in the supplementary information section 1, which shows results for $\sigma^2 = 1$ (extremely smooth) up to $\sigma^2 = 8$ (granular).

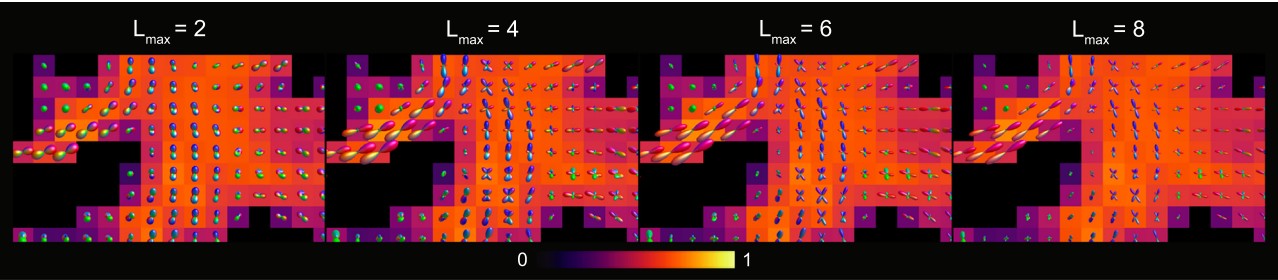

**Fig. 4 | Visualization of the centrum semi-ovale for different spherical harmonics orders on the synthetic datasets.** Different combinations of $l_{max}$ (rows) and datasets (columns) are shown, with parameter map $f_i$ as background. Fiber orientation distributions are scaled for visibility.

**Fig. 5 | Visualization of the centrum semi-ovale for different spherical harmonics orders ($l_{max}$) of the in vivo dataset.** Fiber orientation distributions (FODs) are shown for increasing $l_{max}$ with parameter map $f$ as background. FODs are scaled for visibility.

Fitting time is a critical factor when applying dMRI microstructure modeling, and various efforts have been made to speed up the computationally heavy nonlinear optimization and enable large-scale population studies[19,29,30]. INRs circumvent this through its inherent self-supervised multi-layer perceptron (MLP) structure, which allows for efficient, continuous representation of the parameter space without requiring voxel-wise optimization. The INRs are fit on consumer-grade hardware in around 5 up

to at most 23 minutes for the scenarios tested, much faster than classic NLLS approaches. Using the analytical integration approach decreases training time considerably, possible when excluding negative $b_\Delta$ values in acquisition protocols. Once the INR is fit to the dataset, the inference time is negligible. For example, an $l_{max} = 2$ model with $n_h = 2048$ and $n_p = 5000$ can perform inference at one million coordinates in 2.7 seconds and for $l_{max} = 8$ in 2.8 seconds, including data writing times to and from the GPU. However,

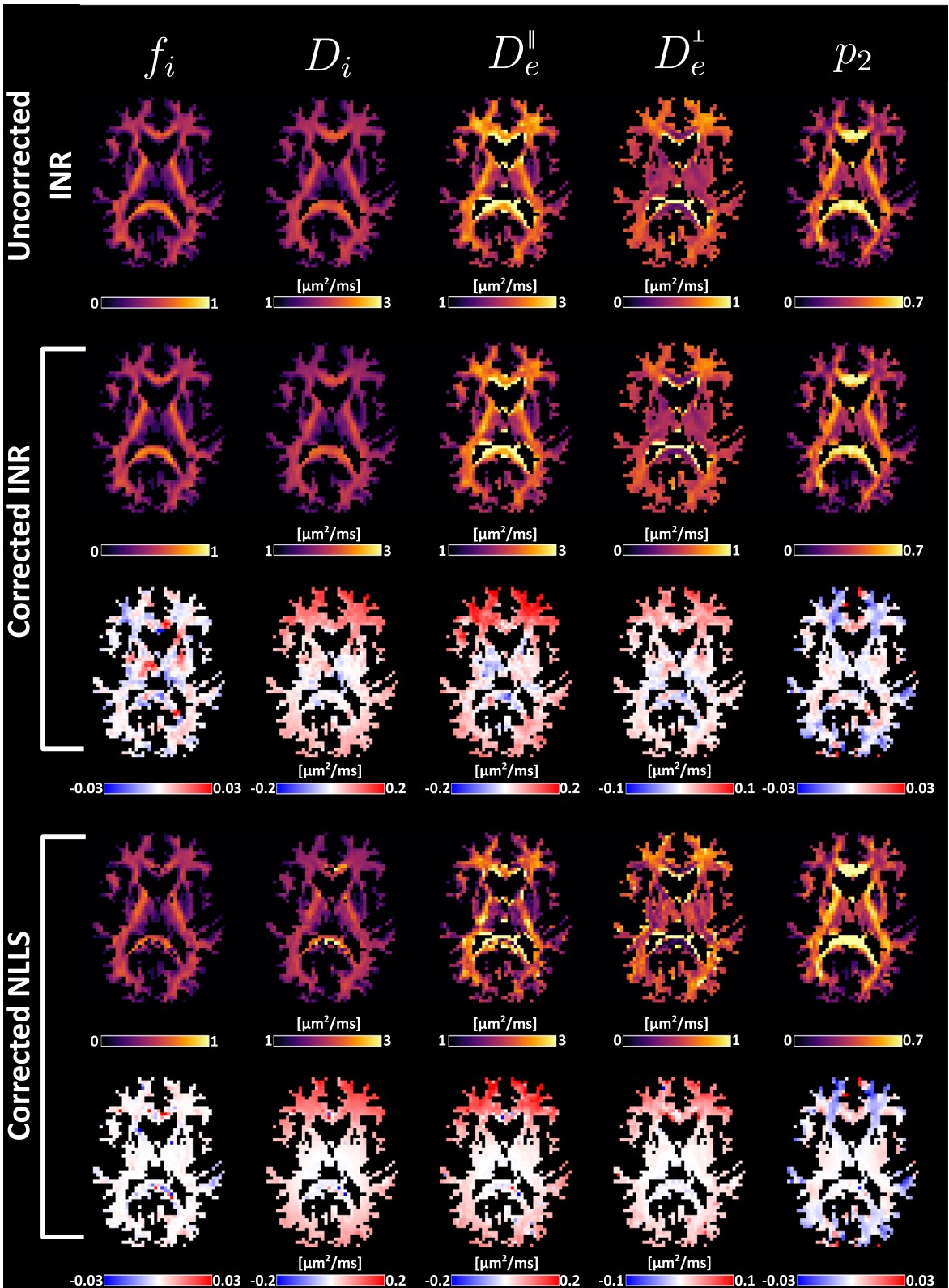

**Fig. 6 | Effect of gradient non-uniformity correction on the estimation of Standard Model parameters.** The top row shows parameter maps without correction. The middle rows show the effect of gradient non-uniformity correction using the INR method. Bottom rows show the effect of gradient non-uniformity correction on NLLS parameter estimation. The difference maps are computed relative to the parameter estimates obtained with the same method, but without applying gradient non-uniformity correction.

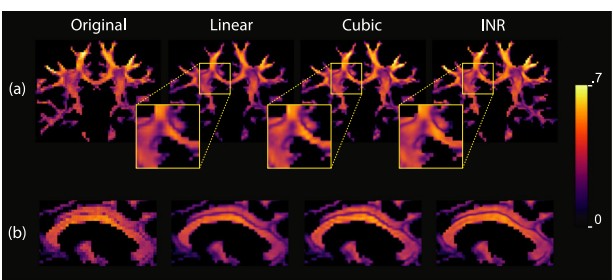

**Fig. 7 | Comparisons between different upsampling methods.** A coronal slice (**a**) and a sagittal slice (**b**) of the $p_2$ parameter map are shown at the original resolution, and upsampled 8x in every dimension using linear interpolation, cubic interpolation, and the INR.

**Table 1 | Table showing fitting times in seconds for the INR on different datasets, N = numerical integration, A = analytical integration**

| $l_{max}$ | 2 | 4 | 6 | 8 |
|---|---|---|---|---|
| Synthetic noiseless (A) | 313 | 448 | 667 | 1097 |
| Synthetic SNR 50 Gaussian (A) | 320 | 462 | 674 | 1100 |
| Synthetic SNR 50 Rician (MSE, A) | 313 | - | - | - |
| Synthetic SNR 50 Rician (Rician loss, A) | 321 | - | - | - |
| In vivo (N) | 390 | 563 | 889 | 1411 |
| In vivo (Rician loss, N) | 391 | - | - | - |
| Gradient non-uniformity correction (N) | 516 | - | - | - |

since INRs require a model to be fit to every individual subject, a supervised learning approach could remain faster for large multi-subject datasets consisting of many subjects with identical acquisition protocols, despite the considerable amount of training time it requires initially (e.g., 83 minutes on GPU, excluding initial NLLS parameter estimations, for the supervised NN method[18]).

The method's ability to incorporate gradient non-uniformity correction in the fitting process provides an advantage over typical supervised methods, for which the training set would be impractically large to capture the spatial variability. This correction is essential as even small non-uniformities can affect parameter maps[28,31–33], and the availability of high-performance gradient coils suffering from significant gradient non-uniformities is increasing[34]. To our knowledge, SMI is the only framework that has incorporated gradient non-uniformity correction into the SM fitting process, in the form of PIPE[35]. This approach uses SVD and linear regression to approximate the exact acquisition settings in each voxel for linear tensor encoding (LTE) acquisitions.

### Limitations of the work

A limitation of using simulated data for evaluation is the variety of possible approaches to generating the ground truth. In this work, the intention was to create a ground truth with structurally smooth characteristics assumed to mimic real brain tissue. However, factors such as voxel size play a role and need to be further investigated. The ground truth generated for the synthetic experiments makes use of SMI for generating the underlying parameter maps. This could potentially bias the parameters to be in a range that favors estimation using SMI. We indeed observed that estimation on the noiseless signal showed optimal performance for SMI and NLLS (see supplementary information Fig. S2). The INR method exhibits lower performance on noiseless signals due to its inability to model voxels individually, indicating that the ground truth is not positively biased with respect to the outputs of the INR method. Nevertheless, we have attempted to reduce biases that would benefit a particular method by smoothing the parameter maps and

using FODs from a different source (MSMT-CSD). Omitting the smoothing still resulted in the highest performance for INR, see supplementary information section 6 Fig. S8. Another limitation related to the ground truth is that the MGH dataset used in this study contained only LTE acquisitions, which may have led to inaccurate parameter estimates (see discussion at the end of this section). We have investigated the impact of other possible sources of severe bias on creating ground truth parameter maps from the MGH dataset, such as the relatively short diffusion time, noise estimation procedure, and included b-values (see supplementary information section 6, Figs. S6–S8). We found similar overall distributions and linear voxel-wise correlations when using longer diffusion times, noise estimate from repeated $b = 0$ smm$^{-2}$ images, and excluding $b = 200$ smm$^{-2}$ and $b > 10.000$ smm$^{-2}$ images. Nevertheless, creating a ground truth that balances capturing anatomical reality while exerting sufficient control remains an important avenue to further explore.

The comparison experiments across methods were conducted using Gaussian noise, which differs from the noise characteristics of in vivo magnitude MRI data, typically following Rician or non-central Chi distributions. However, with appropriate preprocessing and ideally the availability of phase data, the noise can be transformed to approximate a Gaussian distribution[36,37]. This makes the use of Gaussian noise still relevant and consistent with previous work in self-supervised learning for dMRI[38]. The Rician noise experiments reveal biases in the parameter estimates by the INR when using MSE, suggesting that this should be taken into consideration. In this work, we show the promise of correcting for this bias by using a loss function tailored specifically to Rician noise[39].

The INR shows a slight overestimation in $D_e^\parallel$ in the splenium of the corpus callosum in the synthetic experiments. This could be due to the ground truth exhibiting less structural coherence in this part, which is especially apparent in $D_e^\perp$ parameter map. Since the parameters are estimated jointly, this might influence the estimation of $D_e^\parallel$. Potentially, SMI does not suffer from this because it fits the SM voxel-wise and is, therefore, able to produce these combinations of parameters.

The interpretation of the in vivo parameter maps is subjective, as there is no ground truth available. The INR produces more spatially coherent estimates than other methods, showing anatomically plausible structure and physiologically plausible parameter values. This could imply that they more closely resemble the actual underlying tissue, but conclusions should be drawn with caution. For example, compared to the other maps, the INR produces a slightly higher estimate for $D_e^\parallel$ and a slightly lower estimate for $D_e^\perp$. We cannot be certain about which estimate is more accurate. To evaluate the eligibility of the in vivo acquisition protocol to fit SM itself, ground truth simulations with this protocol were performed, which can be found in the supplementary information section 7 (Fig. S9).

Additionally, correcting for gradient non-uniformities has a significant impact on the parameter estimates, yet the accuracy remains to be evaluated, although comparison to NLLS in combination with gradient non-uniformity correction shows similar results. While changes in shape due to gradient non-uniformities are taken into account ($b_\Delta$), a limitation of the current implementation is that it assumes conservation of B-tensor axial symmetry, an assumption that generally does not hold when gradient non-uniformities and non-LTE encodings are considered[35]. The exact impact of this approximation would necessitate implementation of SO(3) convolutions and requires further investigation. Experiment 5 shows that these corrections result in lower estimates for $D_e^\parallel$ which brings $D_e^\parallel$ more in agreement with previous work showing $D_i > D_e^\parallel$ (in gadolinium based contrast experiments[40]) and $D_i \sim 2.3 \ \mu m^2 \ ms^{-1}$ (in experiments with elaborate acquisition protocols using high diffusion planar tensor encoding[41]). Any further inconsistency with values of $D_e^\parallel$ for the in vivo dataset could be caused by the inability of the acquisition protocol to discriminate solution branches as a high b-shell ($b > 5000$ smm$^{-2}$) with LTE is lacking[11,15,42]. To gain more insight into the degeneracies of the INRs estimation and to enable error quantification, calculating the posterior distribution is required[43,44].

## Future work

The presented INR method can be extended in future work. This work has focused on including the minimal number of SM parameters to reduce the complexity of the fitting parameter space and to evaluate the method's performance. Importantly, the framework is fully flexible to fit any biophysical model, by adjusting the forward equation to predict the signal (Fig. 9). For example, the SM implementation can be extended to include relaxation effects, which introduce compartmental $T_2$ as fitting parameters. Further distinction can be made between intra-axonal compartmental $T_2$ and extra-axonal compartmental $T_2$[45,46], which adds two extra fitting parameters. The contribution of free water can also be introduced as a fitting parameter. However, previous work has shown that the impact of this parameter is small except for voxels around the ventricles[15]. Adding this parameter could resolve fitting issues around the ventricles for the in vivo data in experiment 3 where high $D_e^{\parallel}$ and $D_e^{\perp}$ values are found.

The spatial regularization inherent to INRs in the fitting process can be beneficial for other biophysical models. The presented method fits the Standard Model of white matter, which – as the name suggests – is only applicable for white matter. As a result, we applied a white matter mask, and only used the coordinates that lie inside this mask as input to the INR. This implies that for coordinates outside of the mask, and in different tissue types, INR results are not fit (correctly). Implementation of gray matter models (e.g. as in ref. 47) using INRs avoids the need for fitting solely white matter and can provide whole brain parameter maps.

The combination of estimating SM parameters together with FOD SH up to high $l_{max}$ values opens up the possibility to combine the microstructural information of SM-estimates and the directional information of the FOD to do microstructure-informed tractography[25,26,43,48], and future work could extend the model to estimate fiber-direction specific kernels.

When applying this method to a large number of subjects, for example when doing group analysis, fitting times of the INR might become a limiting factor. The duration of the fitting process for self-supervised learning in combination with dMRI models can potentially be considerably shortened when applying transfer learning[49], meta-learning[50], continual learning[51], or hash-encodings[52], which could make on-the-fly fitting of INRs possible. Tractography can also benefit from the continuous representation of the INR in both interpolation computation time and accuracy[53].

The application of microstructural information in clinical studies remains untested in this context, but its potential utility for the diagnosis and assessment of various pathophysiological processes could be explored. The INRs performance remains to be further evaluated in pathology, particularly the effect of spatial regularization on the quantification of small lesions. The encoding frequency variance $\sigma^2$ can be tuned to accommodate higher frequency changes in the signal. The performance of INRs for sparse, clinically feasible acquisition protocols remains to be investigated and represents a direction for future research[38].

## Conclusion

Using INRs to fit the SM provides noise-robust, spatially regularized parameter estimates. FODs of SH orders up to at least eight can be estimated alongside the SM kernel parameters. The self-supervised nature of this approach has advantages over existing (supervised) methods, as it prevents training set bias and allows for explicit correction of gradient non-uniformities, within reasonable estimation times.

## Methods

### Standard Model of white matter

Multiple approaches have been suggested to model white matter dMRI signal as a combination of sticks and anisotropic Gaussian diffusion compartments[9,11,16,46,54–56]. Generalization of this principle without introducing constraints on model parameters has led to a unified framework called the Standard Model of white matter[4,14]. The Standard Model assumes the measured signal $S$ to be described by the convolution of a kernel $\mathcal{K}(b, b_{\Delta}, \boldsymbol{n} \cdot \boldsymbol{u})$ – describing the signal arising from water diffusing within and around a coherent fiber bundle with direction $\boldsymbol{n}$ – with a distribution of fiber populations $\mathcal{P}(\boldsymbol{n})$ on the unit sphere:

$$S(b, b_{\Delta}, \boldsymbol{u}) = S_0 \int_{S^2} \mathcal{K}(b, b_{\Delta}, \boldsymbol{n} \cdot \boldsymbol{u}) \mathcal{P}(\boldsymbol{n}) \, d\boldsymbol{n}. \tag{1}$$

where $b$ is the b-value, $b_{\Delta}$ is the B-tensor shape, $\boldsymbol{u}$ describes the first eigenvector of the B-tensor, and $S_0$ is the signal without diffusion weighting. Our implementation of the SM assumes fiber bundles to consist of two compartments, intra-axonal and extra-axonal, that hold different diffusion characteristics. The signal from an axially symmetric tensor (zeppelin) compartment depends on its axial ($D^{\parallel}$) and perpendicular ($D^{\perp}$) diffusivity and is given by the following relation[57]:

$$\begin{aligned} &\mathcal{K}_{zep}(b, b_{\Delta}, \boldsymbol{n} \cdot \boldsymbol{u}) \\ &= \exp\left[\frac{1}{3}bb_{\Delta}(D^{\parallel} - D^{\perp}) - \frac{1}{3}b(D^{\parallel} + 2D^{\perp}) - bb_{\Delta}(\boldsymbol{n} \cdot \boldsymbol{u})^2(D^{\parallel} - D^{\perp})\right] \end{aligned} \tag{2}$$

The intra-axonal compartment is modeled as a zero-radius stick (i.e. $D^{\perp} = 0$) with $D^{\parallel} = D_i$, while the extra-axonal compartment is modeled as a zeppelin with axial and perpendicular diffusivity $D^{\parallel} = D_e^{\parallel}$ and $D^{\perp} = D_e^{\perp}$, respectively. The fraction of the signal occupied by the intra-axonal compartment is given by $f_i$ (thus setting the fraction of the extra-axonal compartment to $1 - f_i$). Summing over the intra-axonal and extra-axonal signal contributions, results in the following forward equation for the signal:

$$\begin{aligned} S(b, b_{\Delta}, \boldsymbol{u}) = S_0 \cdot &\left[ f_i \int_{S^2} \exp(-bb_{\Delta}(\boldsymbol{n} \cdot \boldsymbol{u})^2 D_i)\mathcal{P}(\boldsymbol{n}) \, d\boldsymbol{n} \cdot \exp\left(\frac{1}{3}bb_{\Delta}D_i - \frac{1}{3}bD_i\right) \right. \\ &\left. + (1 - f_i) \int_{S^2} \exp(-bb_{\Delta}(\boldsymbol{n} \cdot \boldsymbol{u})^2(D_e^{\parallel} - D_e^{\perp}))\mathcal{P}(\boldsymbol{n}) \, d\boldsymbol{n} \cdot \exp\left(\frac{1}{3}bb_{\Delta}(D_e^{\parallel} - D_e^{\perp}) - \frac{1}{3}b(D_e^{\parallel} + 2D_e^{\perp})\right) \right]. \end{aligned} \tag{3}$$

Calculation of the integral can follow two approaches. The first approach leverages an analytical expression for the integral containing a product of Legendre polynomial function and an exponential term. This term arises when projecting $\mathcal{P}(\boldsymbol{n})$ on a SH basis. For a full derivation, see refs. 15,45,58. However, this analytical expression is only valid when $b_{\Delta} \geq 0$ and $D_e^{\parallel} > D_e^{\perp}$ (see ref. 54 for the derivation of this analytical solution). The second approach uses numerical integration to calculate the integral. This approach is able to incorporate negative $b_{\Delta}$ but is computationally more demanding than the analytical approach.

### INR network architecture

The purpose of the INR is to map a coordinate vector $\boldsymbol{x}$ to a desired output vector $\boldsymbol{k}$ which represents the underlying dataset at that coordinate, by passing the coordinate through a neural network $\mathcal{F}_{\Psi} : \boldsymbol{x} \to \boldsymbol{k}$ with weights $\Psi$. In our case we map a 3D-coordinate $\boldsymbol{x} \in \mathbb{R}^3$ to a vector of parameters for the SM kernel and FOD, $\boldsymbol{k} = [D_i, D_e^{\parallel}, D_e^{\perp}, f_i, S_0, p_0^0, ...p_l^m]$ where $p_l^m$ is the coefficient of the SH basis function of order $l$ and phase $m$. The signal is hence projected onto real SH as in ref. 58. The end result is a representation of a (dMRI) dataset of a single subject by a neural network, from which the parameter maps can be inferred at any $\boldsymbol{x}$. A 'dataset' in this manuscript will refer to all dMRI volumes in a single acquisition of a single subject, unless specified otherwise. The implicit neural representation consists of three parts: the spatial encoding, a small MLP, and a number of output layers (Fig. 8). Each of the parts will be discussed in-depth in the upcoming sections.

### Spatial encoding

By encoding the input coordinates to a high-dimensional space before entering them into the model, we can greatly increase the representational power of the INR[59]. We use the Fourier features encoding described by Tancik et al.[59], which was used previously to model MSMT-CSD using INRs[24]. First we scale the coordinates $\boldsymbol{x}$ (maintaining aspect ratio) to lie in

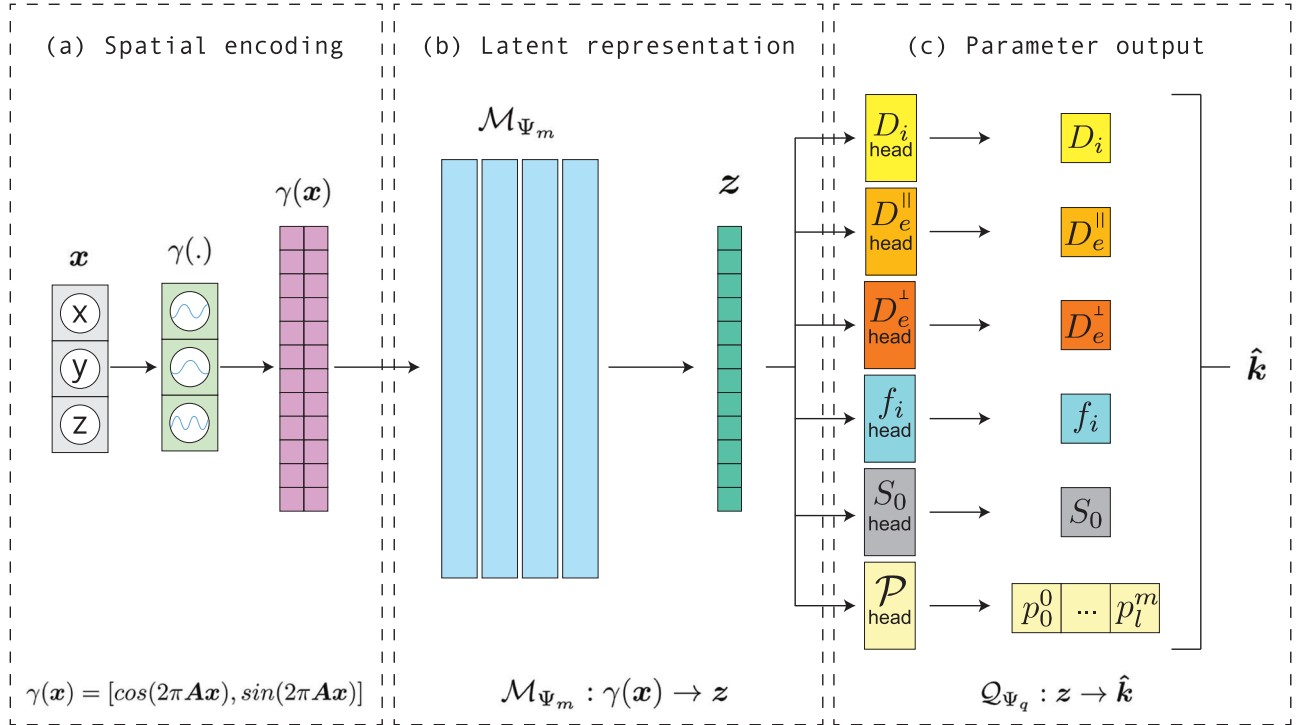

**Fig. 8 | INR network architecture. a** shows how the input coordinates ($x$) are mapped to a higher dimensional frequency space ($\gamma$). **b** These values are then forwarded to the Multi-layer perceptron ($\mathcal{M}_{\Psi_m}$). **c** Output layer ($z$) of the MLP is converted to SM parameters ($\hat{k}$).

range $[-1, 1]^3$, and then apply to following transformation:

$$\gamma(x) = [\cos(2\pi Ax), \sin(2\pi Ax)] \tag{4}$$

where $\gamma(.)$ is the Fourier feature encoding, and $A$ is a size $n_p \times 3$ matrix with values sampled from $\mathcal{N}(0, \sigma^2)$. The number of encodings $n_p$ and the variance $\sigma^2$ are hyperparameters that can be adapted to suit datasets of varying complexity and quality. This process results in an encoded coordinate vector $\gamma(x) \in [-1, 1]^{n_p \times 2}$.

**Multi-layer perceptron**

The MLP $\mathcal{M}_{\Psi_m}$ with weights $\Psi_m$ ($m$ pointing towards the corresponding MLP) is the backbone of INR and is largely responsible for representing the underlying dataset. It consists of four fully-connected layers of equal sizes $n_h$, determined by the complexity of the represented dataset, and ReLU activation functions. The MLP maps the encoded coordinates to some latent vector $\mathcal{M}_{\Psi_m} : \gamma(x) \to z$ with $z \in \mathbb{R}_+^{n_h}$, that serves as an input to the output layers.

**Output layers**

The final part of the INR architecture maps $z$ to a parameter estimate $\hat{k}$ using a separate fully-connected layer, called 'head', for each parameter estimate. For the SM parameters $\hat{D}_i$, $\hat{D}_e$, and $\hat{f}_i$ the heads use a sigmoid activation function scaled to fit physiological ranges (Table 2). For $\hat{S}_0$ a softplus activation function was used, which ensures positivity, without an upper bound[60]. The SH-coefficients of the estimated FOD $\hat{\mathcal{P}}(n)$ require both positive and negative outputs and, therefore, have no activation function. This results in the full output layer of the INR providing the mapping $\mathcal{Q}_{\Psi_q} : z \to \hat{k}$, where $\mathcal{Q}_{\Psi_q}$ are the output layers with weights $\Psi_q$.

**Model fitting**

Given set of $N_m$ (capital $N$ denoting a fixed number, opposed to lower case hyperparameters $n_p$ and $n_h$) measurements $\{(b_i, (b_\Delta)_i, u_i) | i \in 1, ..., N_m\}$, measured at coordinates $x_j \in X$ with $X \subset \mathbb{R}^3$ being the set of all measured coordinates in the dMRI dataset, the estimated signal $\hat{S}(b_i, (b_\Delta)_i, u_i, x_j)$ at

**Table 2 | Table showing the lower (min.) and upper (max.) bounds of the estimated parameters, diffusivities shown in $\mu m^2 ms^{-1}$**

| | $f_i$ | $D_i$ | $D_e^{\parallel}$ | $D_e^{\perp}$ | $S_0$ | $p_l^m$ |
|---|---|---|---|---|---|---|
| min. | 0 | 0 | 0 | 0 | 0 | -inf. |
| max. | 1 | 4 | 4 | 1.5 | +inf. | +inf. |

coordinate $x_j$ is obtained from the model output $\mathcal{F}_\Psi : x \to \hat{k}$ by calculating the estimated kernel $\hat{\mathcal{K}}(b, b_\Delta, n \cdot u)$ using (2) and convolving with $\hat{\mathcal{P}}(n)$ as in (1). We approximate the desired INR $\mathcal{F}_\Psi$ with weights $\Psi := \{\Psi_m, \Psi_q\}$ by finding the weights $\Psi^*$ that minimize the error (MSE or Rician likelihood loss[39], the latter allowing an explicit correction of Rician noise) between the estimated signal $\hat{S}$ and the measured signal $S$:

$$\Psi^* = \underset{\Psi}{\mathrm{argmin}} \frac{1}{|X|} \sum_{x_j \in X} \sum_{i=1}^{N_m} \mathcal{L}(S(b_i, (b_\Delta)_i, u_i, x_j), \hat{S}(b_i, (b_\Delta)_i, u_i, x_j)) + \Lambda_{x_j} \tag{5}$$

where $\mathcal{L}$ is either the MSE or the Rician likelihood loss. We include an additional term $\Lambda_{x_j}$ which is a non-negativity constraint for the FOD at $x_j$, as described by Tournier et al.[10]. The constraint is calculated by sampling the FOD across the spherical domain and adding any negative values as a loss. The full dMRI dataset is used, without a train/test split, as the goal is for the INR to represent the data, not to predict unseen data. The fitting process is shown in Fig. 9.

**Implementation**

The INR is implemented in Python 3.10.10 using PyTorch 2.0.0. An Adam optimizer was used with a learning rate of $10^{-4}$, $\beta_1 = 0.9$, $\beta_2 = 0.999$, $\epsilon = 10^{-8}$, and no weight decay. The hyperparameters were set at $n_p = 5000$, $n_h = 2048$, $\sigma^2 = 3.5$ for the SNR 50 synthetic datasets (see section 'Generation of simulated ground truth data') and the in vivo dataset (see section 'In vivo

data acquisition'), and $\sigma^2 = 2.5$ for the SNR 20 synthetic datasets. More details about the choice of $\sigma^2$ is given in the supplementary information section 1 (Fig. S1). Each INR was fit for 150 epochs on an NVIDIA RTX 4080 GPU with 16GB of VRAM, with a batch size of 500. Visualizations of the model output were created using matplotlib 3.8.0, and MRtrix3 3.0.4[61]. Numerical integration to calculate the integral during training is implemented with Torchquad Simpson function[62].

## Comparisons

The performance of the INR model (referred to as the INR method) is compared to three other SM model fitting methods described below. A supervised machine learning approach using the SMI toolbox[15] (SMI method) with standard settings. The SMI method requires a noise map, which was determined using MP-PCA denoising[63]. A supervised deep learning method (referred to as the supervised NN method introduced in ref. 18) was trained using a combination of synthetic and data-driven parameter samples. Specifically, the training data consisted of 500,000 samples, with 75% allocated for training and 25% for validation. Half of the training samples were generated by uniformly sampling model parameters, while the other half were derived by applying mutations to NLLS estimates obtained from the target dMRI data. The neural network architecture comprised three hidden layers with 150, 80, and 55 neurons. Gaussian noise was added with SNR 50. For a comprehensive description of all training settings, see ref. 18. Finally, an NLLS approach (NLLS method)

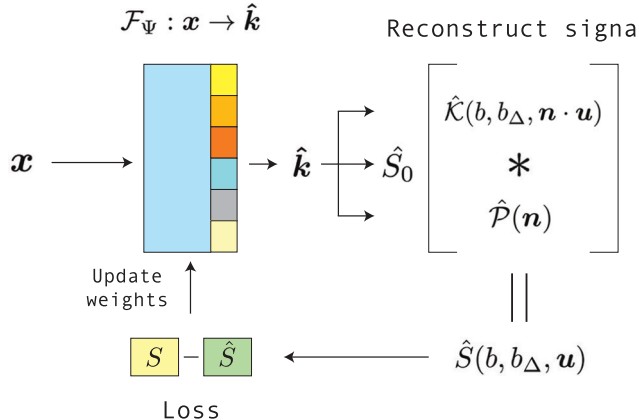

$\mathcal{F}_\Psi : \boldsymbol{x} \rightarrow \hat{\boldsymbol{k}}$

Reconstruct signal

**Fig. 9 | Visualization of the INR fitting process.** The input coordinates $\boldsymbol{x}$ are input in the INR (architecture shown in Fig. 8) and are mapped to a parameter estimate $\hat{\boldsymbol{k}}$. Using $\hat{\boldsymbol{k}}$ the signal estimate $\hat{S}$ is reconstructed following (3). The loss between $\hat{S}$ and the measured signal $S$ is used to update the INR weights.

was implemented with the MATLAB (MathWorks, Natick, MA, USA) optimization toolbox: lsgnonlin with Levenberg-Marquardt algorithm, max 1000 iterations. Two initializations were fitted after which the solution with the lowest residual norm was chosen. Of the above methods, only INR has a positivity constraint implemented for the FOD (see section 'Model fitting').

Fitting performance across the different methods was evaluated using $\rho$ and RMSE on kernel parameters and rotational invariant $p_2 = \sqrt{\frac{4\pi}{5}}\sqrt{\sum_m |p_{2m}|^2}$, where $|p_{2m}|^2$ is the absolute value of the second order, $m$-th phase SH-coeffient.

## Generation of simulated ground truth data

In silico experiments were conducted on simulated data obtained from one brain (subject 011) of the MGH Connectome Diffusion Microstructure Dataset[64]. The dMRI data were acquired on the 3T Connectome MRI scanner (Magnetom CONNECTOM, Siemens Healthineers) at 2mm isotropic resolution. The acquisitions with b = [0, 50, 350, 800, 1500, 2400, 3450, 4750, 6000]smm$^{-2}$ and $\Delta = 19$ms were selected. The lowest 4 b-values were acquired with 32 uniformly distributed diffusion encoding directions, the highest 4 b-values with 64. The dataset also contained 50 $b = 0$ smm$^{-2}$ volumes. More details about the imaging parameters and processing can be found in ref. 65. The SM was fitted with the SMI toolbox[15] to generate a set of realistic SM kernel parameters for $f$, $D_i$, $D_e^{\parallel}$, and $D_e^{\perp}$, using $l_{max} = 4$ and noise bias correction with a sigma map acquired through MP-PCA[63]. Further settings were 2 compartments (intra- and extra-axonal), $10^6$ training samples, and $N_{levels} = 1$. To enhance the smoothness of the kernel maps, anisotropic diffusion filtering was performed using MATLAB's imdiffusefilt function with three iterations ($N = 3$) and minimal connectivity. The SH-coefficients $p_{lm}$ of the FODs were calculated using MSMT-CSD[13] for $l_{max} = [2, 4, 6, 8]$. The simulated signals corresponding to these parameters were calculated from the SM signal equation with a published optimized acquisition protocol[15]: b = [0, 1000, 2000, 8000, 5000, 2000] smm$^{-2}$, number of directions [4, 20, 40, 40, 35, 15], and B-tensor shape $b_\Delta = [1, 1, 1, 1, 0.8, 0]$. Directions were optimized by minimizing the electric potential energy on a hemisphere[66] after which half of the directions were flipped. The image resolution was kept identical to the original dataset at 2 mm isotropic. Finally, Gaussian or Rician noise was added. The standard deviation of the noise distribution was determined by the mean of the $b = 0$ smm$^{-2}$ acquisitions and the SNR (20,50 or $\infty$ on the $b = 0$ smm$^{-2}$ images), resulting in a spatially varying standard deviation. These SNR levels are comparable to (50), or below (20), those investigated in previous work[15,18]. Free water contributions and TE dependence were not considered. Non-white matter voxels are masked out as their influence on the loss value will decrease the performance of parameter estimation on white matter voxels. A white

## Table 3 | In vivo diffusion acquisition scheme and imaging parameters

| $b_\Delta$ | b-value [smm$^{-2}$] | | | | | | | | | | | |
|---|---|---|---|---|---|---|---|---|---|---|---|---|
| | 0 | 100 | 200 | 700 | 1300 | 1450 | 1800 | 2400 | 2700 | 2750 | 3000 | 4000 |
| 0 | 2 | 6 | x | 6 | 6 | x | 6 | 6 | x | x | 6 | x |
| -0.5 | 1 | x | 3 | x | x | 10 | x | x | 3 | 20 | x | 42 |
| 0.5 | 1 | x | 3 | x | x | 10 | x | x | 3 | 20 | x | 42 |
| 1 | 2 | x | 6 | x | x | 20 | x | x | 6 | 40 | x | 84 |
| **Parameter** | | | | | | | | | | | **Value** | |
| Echo Time TE [ms] | | | | | | | | | | | 77 | |
| Repetition time TR [ms] | | | | | | | | | | | 2800 | |
| Voxel size [mm³] | | | | | | | | | | | 3 × 3 × 3 | |
| Field-of-view [mm³] | | | | | | | | | | | 210 × 210 × 45 | |
| Time between diffusion gradients [ms] | | | | | | | | | | | 7.4 | |
| Diffusion gradient duration [ms] | | | | | | | | | | | 30 + 24 | |
| Acceleration | | | | | | | | | | | Partial Fourier 6/8 | |

matter mask was generated with the Freesurfer[67] segmentation, which is included in the MGH dataset. Voxels with $D_e^{\parallel} > D_e^{\perp}$ were also masked out as these represent nonphysical behavior.

## In vivo data acquisition

The study was approved by the Cardiff University School of Psychology Ethics Committee and written informed consent was obtained from the participant in the study. All ethical regulations relevant to human research participants were followed. One healthy volunteer was scanned on a 3T, 300 mT/m Connectom scanner (Siemens Healthineers, Erlangen, Germany). Imaging parameters and diffusion acquisition scheme can be found in Table 3. Gradient waveforms for spherical tensor encoding were optimized using the NOW toolbox[68]. The in vivo data was corrected for Gibbs ringing[69], signal drift[70], motion and eddy current correction[71], susceptibility correction[72], and gradient non-uniformity image distortion and B-matrix correction[28].

## Experiment 1: Quantitative comparison on simulated data

All four fitting methods were compared on simulated data. To mimic realistic conditions, Gaussian noise was added during the simulation of the synthetic dataset (SNR = [20,50]). For this experiment, only $l_{max} = 2$ was considered, which is the highest SH order the supervised NN method can fit. As the used optimized acquisition protocol contains only positive $b_{\Delta}$ values, the SM forward model was calculated following the analytical approach.

## Experiment 2: Rician noise bias

The effect of Rician noise bias was investigated by introducing noise sampled from a Rician distribution (SNR = [20,50]), rather than Gaussian. As in experiment 1, only $l_{max} = 2$ was considered. Parameter estimation was performed using both a standard MSE loss and a Rician likelihood loss[39], and the resulting estimations were compared to the ground truth. The integral in the SM forward model is calculated with the analytical approach.

## Experiment 3: Qualitative comparison on in vivo data

To test the INR on in vivo data, SM parameter estimation was executed on the dataset from section 'In vivo data acquisition' using the Rician likelihood loss. The MPPCA map for the SMI fitting was estimated on the unprocessed data and b-value up to 1300 s/mm². Only $l_{max} = 2$ was considered. As the acquisition protocol contained negative $b_{\Delta}$ values, the SM forward model was calculated following the numerical integration approach. The results are compared to the estimates from the methods in section 'Comparisons'.

## Experiment 4: Estimation of SH order up to $l_{max}$ = 8

The performance of the proposed method to model higher order FODs is investigated by fitting the INR with SH orders of $l_{max} = [2, 4, 6, 8]$, for four different datasets. The noiseless synthetic dataset and the SNR 50 and 20 synthetic data provide insight in the accuracy of FOD estimation in noisy data, while the in vivo dataset qualitatively shows the capability of the INR to estimate higher order FODs on realistic datasets.

## Experiment 5: Effect of gradient non-uniformity correction of SM parameter estimation

The impact of gradient non-uniformities on SM parameter estimation was assessed using in vivo data. For each voxel, the b-value, $b_{\Delta}$, and b-vectors were recalculated to account for scanner-specific gradient deviations following[28]. These corrected effective acquisition parameters were then used to fit the model. This analysis was performed for $l_{max} = 2$. The differences between the corrected and uncorrected parameter estimates were subsequently evaluated. Gradient non-uniformity correction was implemented using MSE loss function and compared to gradient non-uniformity correction with NLLS.

## Experiment 6: Implicit neural representation for spatial interpolation

To obtain more detailed insight into the continuous spatial representation of the dataset provided by an INR, the INR fit on the Gaussian noise, SNR 50 synthetic dataset at $l_{max} = 2$ is sampled at 8x the original resolution in every dimension, resulting in 0.25mm isotropic voxels. The parameter map of $p_2$ was visualized in the coronal and sagittal plane for the original resolution output of the model and the linear, cubic and INR upsampling.

## Statistics and reproducibility

All statistical analyses were conducted using custom Python scripts. Comparisons between ground truth and estimated SM parameters on simulated data, as well as on in vivo data, were performed using Pearson's correlation coefficient and RMSE. The analysis workflow relied on SciPy (v1.16.1), sklearn (1.7.1) and NumPy (v2.3.2). No inter-subject statistical analyses were carried out.

## Reporting summary

Further information on research design is available in the Nature Portfolio Reporting Summary linked to this article.

## Data availability

The MGH Connectome Diffusion Microstructure Dataset[65] is publicly available. The synthetic datasets are available at Zenodo (DOI: 10.5281/zenodo.17092773[73]). The in vivo dataset can be made available upon reasonable request and by signing a data sharing agreement with Cardiff University.

## Code availability

The code used to generate the results in this paper is publicly available through GitHub at: https://github.com/tomhend/Standard_model_INR.

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

## Acknowledgements
We thank Dennis Klomp for input to the manuscript, and Bram Kraaijeveld for
the help creating graphics. CMWT is supported by a Vidi grant (21299) from
the Dutch Research Council (NWO) and a Sir Henry Wellcome Fellowship
(215944/Z/19/Z). Data were provided [in part] by the Human Connectome
Project, MGH-USC Consortium (Principal Investigators: Bruce R. Rosen,
Arthur W. Toga and Van Wedeen; U01MH093765) funded by the NIH
Blueprint Initiative for Neuroscience Research grant; the National Institutes
of Health grant P41EB015896; and the Instrumentation Grants
S10RR023043, 1S10RR023401, 1S10RR019307. This research was funded
in whole, or in part, by the Wellcome Trust [215944]. For the purpose of Open
Access, the author has applied a CC BY public copyright license to any
Author Accepted Manuscript version arising from this submission.

## Author contributions
Conception: T.H., G.A., A.V., M.C., C.T.; dataset generation: T.H., G.A., C.T.;
analysis and interpretation of data: T.H., G.A., E.V., A.V., M.C., C.T.; drafting
manuscript: T.H., G.A., M.C., C.T.; revising manuscript: T.H., G.A., E.V., A.V.,
M.C., C.T.; funding: A.V., M.C., C.T.; supervision: E.V., A.V., M.C., C.T. Every
author has approved the submitted version, and is personally accountable
for their own contributions.

## Competing interests
The authors declare no competing interests.

## Additional information
**Supplementary information** The online version contains
supplementary material available at

Tom Hendriks.

**Peer review information** *Communications Biology* thanks Jon Haitz
Legarreta, Benjamin Towle, and the other anonymous reviewer(s) for their
contribution to the peer review of this work. Primary Handling Editors: Sahar
Ahmad and Jasmine Pan. A peer review file is available.

**Publisher's note** Springer Nature remains neutral with regard to
jurisdictional claims in published maps and institutional affiliations.

