## [Transparent Peer Review file · Communications Biology]

Implicit neural representations for accurate estimation of the standard model of white matter

Corresponding Author: Mr Tom Hendriks

Version 0:

Reviewer comments:

Reviewer #1

(Remarks to the Author)

Authors propose to use implicit neural representations (INRs) to compute the standard model (SM) parameters for white matter tissue employing diffusion MRI (dMRI) data. The present work builds upon previous work by the authors (Hendriks et al. Imaging Neuroscience, Volume 3, 2025), where the exact same approach and neural network architecture (up to the output heads) is used to compute the spherical harmonic coefficients on white matter tissue and volume fractions for gray matter and cerebrospinal fluid. In the present work, the input data positional coordinates are mapped to some higher frequency domain through Fourier encoding to finally train a neural network to estimate the SM parameters. The training is fully unsupervised, as the SM parameters are fed to a forward model to compute the dMRI signal which drives the optimization. A number of experiments are presented on simulated and in vivo data in order to study the impact of their method. These include varying the noise level, using Rician noise, or simulating gradient inhomogeneities. Compared to conventional or other supervised machine learning and neural network approaches, authors show that their method outperforms them in terms of the error and correlation with respect to the known ground truth. Graphic results provide a qualitative sense of the variance and bias of each particular method. Authors claim that their method could be used to provide SM estimates that are robust to noise, gradient inhomogeneities, and within reasonable inference time.

The experimental design is limited to the case of a single brain dMRI dataset, including simulated and in vivo data. Although the evidence presented in terms of results shows that INRs improves the compared methods, overall, the training process of the neural network remains unclear, and the generalization to $N > 1$ and other acquisition settings remains to be demonstrated.

Thus, although the body of work presented holds promise, a number of concerns and questions exist that would need to be addressed.

Major comments

1. Authors mention the long/extensive acquisitions in the abstract as a challenge for computing the SM parameters, but do not later (e.g. in the introduction) develop why long acquisitions are not ideal (even if evident). They also mention "INRs demonstrate potential to improve on voxel-based methods to estimate parameters, especially in (...) sparsely sampled datasets" (line 32). However, both datasets used in the experiments are issued by such extensive acquisitions. Thus, it is unclear how the method applies to data that has not been acquired (or simulated) using a dense sampling of b-values, directions and $b_{\{\delta\}}$ values. In order to support the claim that an INR does not require training data from extensive acquisitions, it would be necessary to create a subsampled dataset (directions, shells, and $b_{\{\delta\}}$) and show that training on subsampled data provides accurate estimates wrt those on full data.

2. It is unclear from the introduction what the novelty or originality of this work is with respect to previous scientific literature that have employed INRs to model dMRI data, including authors' previous work (Imaging Neuroscience, vol.3 2025). In that work, authors used the same exact approach (network architecture, Fourier encoding, unsupervised training) to compute SH coefficients (which are also computed in the present manuscript, besides the SM parameters). So readers would benefit from a clearer motivation of the work and distinction with respect to previous work.

3. Authors mention that "(The SM) issues become even more prominent at high noise levels" (line 15) and "INRs demonstrate potential to improve on voxel-based methods to estimate parameters, especially in more noisy (...) datasets" (line 32). However, in practice, results for a single noisy setting (SNR 20, SNR 50 being closer to the noiseless side) are presented. A larger number of low SNR values would be required to demonstrate that INRs provide consistent improvements on low SNR values.

4. The sentence "It also does not require uniform acquisition parameters across voxels, making it capable of explicitly correcting for gradient non-uniformities (...)" is misleading. Unless the shimming completely removes these, gradient inhomogeneities are likely present on any acquisition, and more prominent at stronger field strengths. These can be minimized when processing the diffusion data. Also, these variations are not deliberate; however, the first part of the sentence suggests that INR can perform well on settings where each voxel is deliberately acquired with different acquisition parameters (e.g. b-values, b_{Δ}). Given that acquiring each voxel with different parameters has already been proposed in the literature, in order to support the claim that INRs are able to model data purportedly acquired with different parameters, an experiment that demonstrates it would be needed.

5. It is unclear how the training/testing is performed for the INR model. What is the training and testing data used? How is the data split into training and testing? Since $N=1$ in the work, it is unclear how the INR was trained.

6. In section 4.1 it is mentioned that "the supervised NN method (...) can be significantly more time-consuming due to its reliance on NLLS for generating part of the training data and the need to retrain the model for each specific acquisition protocol.". However, authors do not demonstrate the performance of INR when applied to another acquisition protocol, so a new experiment needs to be added to demonstrate how well their method performs compared to the supervised NN when being applied to another acquisition protocol on which INR has not been trained. This is related to the passage "INRs require a model to be fit to every individual dataset." (line 270). When using a neural network, the expectation is for it to be trained once on some data and then be applied many times to different data. So it is unclear what the gain of using an INR is in this context. Also, if it has to be retrained on every individual dataset, it is unclear how the learning process (training/testing splits, etc.) would proceed.

7. In Table D.4 (Appendix D) results for synthetic noiseless and SNR 50 data are shown, and it is argued that INR is robust to noise when computing parameter maps for different l_{\max} values. However, SNR 50 is arguably not a noisy setting. Results with SNR 20 (and potentially other low signal values) would be required to demonstrate this.

8. Figure 1 shows the output values computed by INR, including the diffusivities, the compartment fractions, S_0 and S_H coefficients. However, it is mentioned that (section 2.5) the p_2 rotational invariant will be used to demonstrate model performance, and such is the case across the figures in the manuscript. However, tables D.4 and D.5 employ S_0 instead of p_2 . What is the reason for such change?

9. When authors mention the inference time (section 4.1, line 269), they provide an example with $l_{\max} 2$: although $l_{\max} 2$ is heavily used across their experiments due to limitations of the models they compare to, it would be informative to measure the time for other l_{\max} values enabled by common modern acquisition settings (e.g. more than 15 directions at least, and maybe 30, even on a clinical setting). What is the inference time for e.g. $l_{\max} 4, 6, \text{ or } 8$? Is the increase linear as the number of S_H coefficients that need to be estimated increases?

10. In section 4.1 authors mention "However, since INRs require a model to be fit to every individual dataset, supervised inference remains faster for datasets consisting of many acquisition parameter settings". What do authors mean by "every individual dataset"? What is the meaning of "dataset" in this context (i.e. does this mean data composed of multiple participants, or data containing multiple acquisition parameters)? Given that $N=1$ in the manuscript, How does the authors' method work when there is multiple subjects? This question is related to the previous questions about the training/testing splits of INR (questions 5, 6).

Minor comments

1. The abstract contains some missing punctuation marks/whitespaces after punctuation marks or excess of these. Please fix those, e.g.

"(...) existing methodsResults demonstrate (...)"

"(...) plausibly in a continuous way.The INR is (...)"

"(...) achieves fast inference , is robust to (...)"

"(...) corrections.The combination (...)"

2. Line 6: Why is "B" in "B-tensor" capitalized? The rest of the diffusion sensitization weighing notations use the common, lowercase "b". Applies to line 149 as well.

3. Line 8: "Alzheimer disease": "Alzheimer's disease".

4. Line 52: "standard model" is not capitalized in here whereas it was in previous mentions to it. Adopting a single case would be more consistent.

5. Line 56. Specify what "n" (bold n) is in equation (1), even if it is evident.
6. The notation around line 61 is misleading: why do authors assign (from the notation) the entire diffusivity of the extra-axonal compartment to the parallel component (i.e. $D_{\text{parallel}} = D_{\text{e}}$), even if they model the extra-axonal compartment as a symmetric tensor having both parallel and perpendicular components? Upper/lowerscripts should be used at least to avoid making the notation confusing.
7. A revision of equations 2 and 3 is necessary: equation (2) has a missing closing bracket in the second operand, which seems to have been mistakenly placed at the end. In equation 3, the first operand is missing the term b, and probably the factor.
8. Please, use a single line for equation (2) to make it easier to read.
9. Please, use the same term to refer to the first block of the network: "spatial encoding" vs. "positional encoding". This applies to the appearances in the text and the name of section 2.2.1.
10. What do authors mean by "signal + coordinates" in the Figure 1 caption? According to the text and the operation of an INR, only the coordinates are used as the input data. If "signal" refers to the additional data used at the input in Experiment 5, please make it explicit.
11. Add the min/max values corresponding to the SH coefficients to Table 1, even if it is to say that they can be $-\infty$, $+\infty$ (or -1, +1).
12. Authors mention in the Implementation section (line 113) that they used "default settings otherwise": what do "default" settings mean in this context? What are those parameters?
13. Please, specify what I_{max} was used in "Experiment 2: Rician noise bias".
14. How were the SM values computed for the Cardiff-acquired data to drive the optimization of the neural network?
15. In section 2.8.5 authors mention that their neural network receives additional parameters to model the gradient inhomogeneities. These parameters should probably be introduced in the methods, as this means that the neural network not only receives the x,y,z vector as the input, but also additional parameters.
16. The resolution of the simulated data is not specified: besides being informative, it seems relevant for "Experiment 6: Implicit neural representation for spatial interpolation". Readers can only do the reverse computation, but it would be good to save this task to them.
17. How do authors explain the effect that INR exhibits some overestimation of D_{e} in the splenium (section 3.1) or again the differences in the splenium and other structures in section 3.3?
18. How do authors explain that when no noise is added, INR does not perform as well (section 3.1)?
19. The Figure 7 caption says "Difference maps are with respect to prediction with MSE and without gradient non-uniformity correction (top row)". Although the reader can assume that the "Corrected+MSE" was computed wrt the "uncorrected" version and the "Corrected+Rician" wrt to the "Corrected+MSE", please make this explicit; if the computations were not these, please be more clear in the caption.
20. In section 4.3 authors mention "When sampling outside of the white matter (as shown around the edges in Figure 8), the INR produces inaccurate results.". It is unclear what authors refer to as discrepancies are not apparent in the figure. If there are, these should be mentioned in the results and pointed with an arrow to then discuss the issues/causes/hypotheses/mitigation measures in section 4.3.
21. Please, revise the missing letters in author names and title in reference 55.
22. Section 4.1 and Appendix B. Do not use "Method 1", "method 2", "Method 3", etc. Please, name the methods as you have done throughout the rest of the article.
23. Appendix B. Line 536: "(...) where Gaussian noise is considered with SNR 20". It should be "SNR 50" according to the text in section 3.1, the caption and the fit of the values shown in the figure.
24. Please, adopt a consistent naming for the "noiseless" data: noiseless: 7 apps; clean: 2 apps. Sticking to one would make the read easier.
25. Lines 249 and 252: "The quantitative experiments on synthetic data show how the proposed method outperforms existing methods on noisy data." and "At higher noise levels (SNR = 20), the INR method clearly outperforms all other methods (see Fig. 3)". These sentences convey the same message/one is a reworded version of the other, with a passage about less noisy data in between. Please, reformat the passage so as to avoid the fragmentation.

26. The parameter sorting across tables and figures (columns) is not consistent: Table 1, figures and Tables D.4 and D5 use all a different sorting for the columns (figures are consistent among themselves, and table D.4 and D.5 are consistent between themselves). Please, sort the columns so that the same sorting is used across all tables and figures.

27. In section 4.1 authors mention "However, since INRs require a model to be fit to every individual dataset, supervised inference remains faster for datasets consisting of many acquisition parameter settings". The term "supervised inference" is uncommon and unclear. Do authors mean "supervised training"?

28. Line 286: The sentence "(...), we have attempted to reduce biases towards the estimation for creating the ground truth by smoothing the parameter maps and (...)" is difficult to read/is not well understood (especially, the part "towards the estimation for creating the ground truth"). Please, re-word.

29. In section 4.2 authors state "Additionally, correcting for gradient non-uniformities has a significant impact on the parameter estimates, yet the accuracy remains to be evaluated. Both experiments in section 3.2 and 3.5 show (...)". The results in section 3.2 are not about gradient non-uniformities. Is this a typo?

Reviewer #2

(Remarks to the Author)

Summary

This paper presents a deep learning-based method for signal estimation, grounded in the Standard Model of white matter. Their work is inspired by the effectiveness of implicit neural representations (INR)--a more general deep learning technique for representing a signal, e.g. an image, as a continuous function (i.e. rather than discrete pixels). They evaluate their method across both synthetic and in-vivo datasets. They compare their approach to 3 other methods and find their INR approach delivers broadly superior results.

Strengths

-S1 Their approach requires only the signal to reconstruct at a particular location. The SM parameters are learned implicitly as a sort of interpretable latent variable. Due to the long latency of baseline methods, this approach is generally much faster including train + inference time.

-S2 INR, unlike voxel-based approaches, is resolution agnostic. Theoretically, the model could make predictions at inter-voxel locations, enhancing its downstream use-cases.

-S3 The proposed method outperforms all three baselines, particularly at higher noise regimes relative to supervised approaches, suggesting their approach may be more robust.

-S4 The method is highly extensible to learning any arbitrary parameters provided their relationship with the signal can be modelled in a differentiable way.

-S5 The code is made publicly available, facilitating reproducibility of the results.

Weaknesses

-W1 As the claims of the paper are not only that INRs are more efficient, but also more accurate than supervised methods, it would be useful to see an ablation on where these performance gains are coming from. Specifically, applying some additional supervision via an auxiliary loss function to predicting the intermediate SM parameters. As the supervised NN baseline does not use an INR framework, this ablation would provide a better apples-to-apples comparison.

-W2 The terminology around 'unsupervised' could be refined. INR is not generally considered to be an unsupervised technique as the signal is a label. I understand it could be considered 'unsupervised' with respect to the SM parameters, but the terminology could be confusing as the proposed technique is closer to indirect/weak supervision.

Other Comments & Typos

42-43 'Moreover, because this is an unsupervised method, it does not rely on large training datasets as supervised methods do'. This claim needs support--generally supervised methods are more sample efficient than unsupervised ones in deep learning.

Abstract '...way.The INR ...' there appears to be no space before 'The', unless its just a latex artifact.

Reviewer #3

(Remarks to the Author)

In the present work, the authors propose the use of implicit neural representations (INRs) to estimate the parameters for the Standard Model (SM) of diffusion in white matter (WM) from diffusion MRI data. Authors have successfully used INRs previously to solve the fiber orientation distributions only, but now they extended their technique to estimate the full set of SM parameters. They argue that the proposed approach improves over existing SM parameter estimation methods as it uses spatial regularization, has the ability to explicitly correct gradient non-linearities, and avoids biases from training data due to its unsupervised nature. They compare their method with two state of the art techniques: standard model imaging (SMI), and

a supervised deep learning method (Supervised NN), showing better results, specially at lower SNR.

The contributions in the method proposed by the authors: spatial regularization, correction of gradient non-linearities and non-biases because of training data, are important and necessary upgrades for SM parameter estimation, and the authors did a good job describing their approach and its advantages in the article. I was impressed on how good their results are for the synthetic data evaluation, in comparison with the other methods. However, I still have a few questions/comments I'd ask the authors to respond.

1

I have some doubts regarding the procedure for the simulated ground truth (GT) data creation. The MGH Connectome Diffusion Microstructure Dataset consist of scans from 26 subjects, in which 7 of them were rescanned, is there any strong rationale for using the scan from only one subject (not even rescanned) for the GT creation? Why not using the scans/rescans from all the subjects? It's my understanding that although these scans were preprocessed, they were not denoised. Then, SNR can be highly increased by using the average from all subjects (either by averaging the dwi's or averaging the resulting parameters, after warping them into a common space), then, it would be probably fine to use NLLS (with multiple initializations), for the computation of the GT parameters, which probably will be less biased than the strange combination of SMI parameters plus MSMT-CSD FOD to generate the synthetic signals. Also, spatial coherence will be enhanced naturally on the averaged cohort instead of artificially applying smoothing on the kernel maps, which may give unfair advantage to the INRs method.

Also, while using an optimized protocol for the simulated GT data was great, it could also be good to report how the methods behave with the protocol of the real in vivo data used in this study, as it does not contain any high b-value ($b \geq 6000$). This could evaluate the biases of the methods due to a not optimal protocol.

Finally, SM is valid for long times, the MGH dataset contains data from two diffusion times, $t=19$ and $t=49$, authors used $t=19$, but why not use the data with the longest time? This could be more appropriate and less biased for SM, and contains enough data for the parameter estimation, even if $b=200$ is excluded (because it contains considerable free water signal, which is not considered in the fitted model) and even maybe excluding $b>10000$ (if SNR is too low).

2

I also have questions regarding the use of the MPPCA method to compute the noise map. As the authors mentioned SMI needs a noise map, which is used to regularize the estimation according to the SNR of the fitted signals (if I understand correctly, although it does not regularize spatially, its function could be somehow analog to the function of hyperparameter σ of the INRs method in the sense of how smooth the resulting parameters maps could be). Well, as mentioned in the previous point, the dataset used for the GT creation was preprocessed but not denoised, then using MPPCA to estimate the noise map it's probably not correct as there are correlations in the noise of the data which violates the assumptions of the MPPCA method. Then, MPPCA could very likely underestimate the noise map. If you compute the data SNR by dividing (b_0 map)/(noise map), which is this value in WM on average? According to the owners of the data, it should be around $SNR=23$. If this value, is much higher, then SMI parameter maps may be less smooth than they could be. Hence, this could be another reason to change the approach for the creation of the GT as I mentioned in the previous point. Also, if using this MPPCA noise map to correct the Rician bias, then the bias not be fully corrected. It may be better to compute σ from the 50 b_0 s in each scan, or use the 14 subjects (7 with rescan) with real-valued data in the MGH dataset.

3

Regarding the estimation using non-linear least squares (NLLS), when working on noisy data, its good practice to use multiple initializations (and then keeping the best result), as the degeneracy of the SM estimation problem, in which there are multiple optima, is well known. Was this multiple initialization approach used for NLLS estimation? The manuscript mentions that the Levenberg-Marquardt algorithm was used with max 1000 iterations, but it does not seem this refers to 1000 initializations, is it?

4

For the evaluation of the upsampling capabilities of the proposed method, 8x of the original resolution, authors compare with simple 3d interpolation methods, which although useful for simple tasks, are probably not be the best to perform the comparison. Why not compare against more appropriate upsampling methods from literature? I can think for example on the Non-local MRI upsampling (Manjon et al. Medical Image Analysis. 2010), with codes available in the webpage of the first author. Although, this method could be too old already, maybe there are newer available method for upsampling.

5

Similarly, for the evaluation of the method capabilities to correct gradient non-uniformities during the parameter estimation (then not requiring uniform protocol across voxels, which is an important problem for high performance scanners such as MAGNUS and Connectome), the authors only compare their method with and without this correction. Why not compare against other methods from literature that also correct gradient non-uniformities? The authors cite one of them in reference 6 (same authors as the SMI method), its preprint version provide the code for the method, could the authors compare against this method and maybe other ones from literature.

6

Regarding the in vivo data in section 2.7, could you please describe in the manuscript the preprocessing steps? Preprocessing data with combined b, beta is not trivial, for example, did you used eddy? If so, how did you handled having same b-values with different betas, and the shells with very few directions (i.e. 3, 6, etc.)?

In Figure 6, could the ROI for the in vivo data match better the ROI for the synthetic data? At first impression it seems that for the in vivo data there are missing crossing fibers in comparison with the synthetic data, but at a closer look they seem to not be positioned at the same level, and I assume the intention of the authors was for them to match.

Minors:

In many cases, such as in the discussion and appendix B, authors mention method 1, method 2, method 3, never defined, instead of calling the methods by their name.

While equation (1) is correct, why not writing it in integral form, as in most literature about Standard Model?

Check for typos in the manuscript, just the abstract contains at least three typos, two in which there is no space between the period at the end of the sentence and the following letter, and another one in which besides the lack of the space there is not period at all, it says "existing methodsResults".

Version 1:

Reviewer comments:

Reviewer #1

(Remarks to the Author)

In the revised version of the manuscript authors have provided answers to the questions raised in the previous review, and have modified passages of the text accordingly, and have added error computation results in the appendix section 7 (tables 1 and 2; formerly tables D4 and D5). It is unclear whether the results they have added as supplementary materials in its section 4 answer adequately to the questions raised. Also, the choice of the single low SNR value they have used to demonstrate their method is still not adequately justified.

Major comments

1. Authors have split and renamed the supplementary material sections, which has made keeping track of changes a lot harder. Additionally, there is no color distinction between new content and previous content in the supplementary materials, including new supplementary material sections. Finally, they have used the same numbering scheme (e.g. without any prefix, using letters exclusively, etc.) for supplementary sections, figures and tables as those used in the main section. All this results in a complex task of checking the references within the rebuttal and the revised version of the manuscript. Should there be another round of reviews, please use a specific numbering for the supplementary material sections, figures and tables, and color new section titles, figure and table names and captions with the color chosen for the new text.

2. Related to R1.C1, authors have still not shown how the protocol generalizes to sparsely sampled datasets, and still keep the argument that motivated the question ("INRs demonstrate potential to improve (...) in sparsely sampled datasets."). They explicitly say "Although not a further reduced protocol, these experiments show generalizability to other acquisition protocols". So, reading the description of section 4 of the appendix related to this experiment, it becomes apparent that the acquisition protocol (table 2) is still not what we would call sparse. To be clear, it is informative for the community/readership the sampling scheme where your method results starts to degrade. Similarly, authors chose SNR 50 to demonstrate results, which again may not be considered a low SNR setting. In that response authors say that studying sparsely acquired datasets is out of the scope of the work: if such is the case, the passage mentioning the potential of INRs to improve estimation on sparse acquisitions should be removed and be discussed.

3. Related to R1.C3, the SNR 25-100 range that authors employ to justify that SNR 20 and 50 are representative of noisy settings is still not convincing enough or not backed. Can authors provide references to works that clearly state that the SNR below 25 in DWI is extremely rare?

4. Related to R1.C7, reviewers cannot know what Figure R2 and Figure R3 refer to, as these names do not exist in the revised version of the manuscript.

5. Related to R1.C8, similar to the previous comment and comment 1, tables D.4 and D.5 are no longer present in the revised manuscript. In this case, given that they are the only tables in the supplementary materials, the reviewer has been able to locate them, but future rebuttals and revised versions should avoid these issues.

6. Related to R1.C9, the question asked referred to the inference time, not the fitting time: authors have pointed to Table 3, which is about fitting. Thus, the question remains to be answered.

Minor comments

1. Related to C1.C12, then the two appearances of the term "B-tensor" in line 58 of the revised manuscript should also be

capitalized.

2. Related to R1.C16, even if the worded description in the rebuttal is clear, the notation is still not and the passage in the manuscript is still misleading. Please, make the necessary changes so that the notation and meaning is not misleading.
3. Related to R1.C17, the equations are still not right: for Eq 2: please check Tax et al. Neuroimage 236 (2021), equation A.3: the brackets are either not OK, or the b multiplier in the right-most operand is now redundant; for Eq 3: the exponent of the term $\exp(1/3 b D_{\{i\}} - 1/3 b D_{\{i\}})$ is now zero, which was not the case previously, and $b_{\{\delta\}}$ has disappeared from the right-most term. Please fix this or explain if there was an error in the previous version of the manuscript.
4. Related to R1.C27, a form of the response should be added to the discussion.
5. Related to R1.C28, authors answer that "The INR could get closer to the ground truth in noiseless scenarios by increasing sigma and the MLP size.". Can this be demonstrated with an experiment? Otherwise, the argument is void.
6. Related to R1.C29, in response to the question, authors say that they have changed the figure caption. Unfortunately, it is not the case. Please, change the caption to avoid confusion.
7. In response to comment C30 (R1.C30), by removing the referenced to the figure, authors have left a statement that is not supported by any numerical or graphical result, as they only show WM data, which is not helpful. Please, provide references or elaborate on the statement so that readers that have not run any INR model can understand exactly the limitation that they are referring to.
8. Line 149: there is a typo in the passage "(...) and 1 Nlevels".
9. Line 213: "Parameter estimation on simulated data without noise and with SNR = 50 can be found in the supplementary information (...)". Please, point to the specific section/figure.
10. Line 235: "This holds true for all kernel parameter estimates, which is shown in further detail in the supplementary information section 6.". Did authors mean section 7 instead of section 6? Related to major comment 1, please, add a prefix (e.g. S) or use letters exclusively to number the supplementary material sections so that it becomes unequivocal when authors are referring to them. Same goes to the figures and tables.
11. It looks like section 4/figure 5 of supplementary materials is not cross-referenced in the manuscript. Or it is not evident, at least.
12. The section 3.6 text should appear before the figure that it explains (Figure 9).
13. Line 269: "Parameter estimates for FOD up to SH order eight can be provided alongside (...)". The sentence suggests that parameters of larger SH orders cannot be provided. Although in their experiments authors employ SH orders up to 8, this should not mean that higher order coefficients cannot be provided. Please, reword the passage.
14. Line 284: I_{\max} : Please, use the same notation used throughout the rest of the manuscript, making "max" be a subscript.
15. Line 305: the sentence "The lower performance of the INR method on simulated data without noise arises from its inability to capture the input data perfectly, as it does not model individual voxels, unlike the other methods." is misplaced as it is written between two sentences that are related to each other. Please, move the sentence to a better location in the passage.
16. Line 311: "We have investigated the impact of other possible sources of severe bias on creating ground truth parameter maps from the MGH dataset, such as the relatively short diffusion time, noise estimation procedure, and included b-values (results not shown)". Please show those results as supplementary materials.
17. Line 315: please, fix the typo in "(...) and $b_{\delta} 10.000 \text{ s/mm}^2$ ".
18. Line 333: the sentence ending period is missing in "(...) encodings are considered (7) The exact (...)".

Reviewer #2

(Remarks to the Author)

Summary

This paper presents a deep learning-based method for signal estimation, grounded in the Standard Model of white matter. Their work is inspired by the effectiveness of implicit neural representations (INR)--a more general deep learning technique for representing a signal, e.g. an image, as a continuous function (i.e. rather than discrete pixels). They evaluate their method across both synthetic and in-vivo datasets. They compare their approach to 3 other methods and find their INR approach delivers broadly superior results.

I would like to thank the authors for their extensive work to reply to and make corrections based on the feedback from all

reviewers. Below I include a response to the changes made as well as additional comments on other items to improve. I believe the authors have largely addressed my points, with some caveats on comment 2. Beyond this, I have included additional comments that are largely areas in the paper that would benefit from additional clarification, as well as a few minor typographical issues.

Comments

- 1) I thank the authors for updating the usage of unsupervised to self-supervised. I believe this is now more in-line with standardised usage, while still emphasising the novelty of the approach compared to previous supervised techniques.
- 2) I thank the authors also for their discussion around the effects of SNR and how INR appears to show greater robustness in lower SNR regimes. However, as I see it, there are two core innovations in this paper: the application of a continuous, rather than discrete representation over voxels (INR) and the self-supervised approach that discovers the SM parameters. From here, it is not obvious which is providing the core benefit here. If we imagine self-supervised INR as one quadrant, then supervised INR and voxel-based self-supervised would be other possible ways to approach the problem. The authors indicate in their reply that a supervised INR would be impractical due to the one model per-dataset fitting. However, even if both these quadrants are impractical it would be useful to clarify this in the paper, as otherwise it isn't clear to me at least why the baseline supervised NN from (13), without being deeply familiar with that paper, couldn't also use INR.
- 3) Based on R1 C6, I think it would be useful to clarify in the paper the lack of train / test split in the paper. As far as I can see, INRs do typically have a train / test split (e.g. Mildenhall et al., 2020. NeRF: Representing Scenes as Neural Radiance Fields for View Synthesis). I understand why the split may not matter as much in this case due to the lack of direct supervision on the SM parameters, but may be worth clarifying in the paper.
- 4) Related to this, did the authors observe any impact to the SM parameters from over / underfitting to the dataset? Is there a way to evaluate the SM parameters mid-training run to implement early stopping?
- 5) Line 25: I think it would help to be specific what is being supervised or not in the case of self-supervised, and how the self-supervised case only relies on the signal.
- 6) Line 32: INR is described as different from voxel-based methods, but it isn't clear to anyone unfamiliar with INR why it is different. This is further made confusing by the reference to 'per voxel' in line 46.
- 7) There is some inconsistency around use of dashes throughout the paper, e.g. in-vivo versus in vivo, noise-robustness versus noise robustness.
- 8) Line 49: It is unclear why the proposed approach is not biased by the training data. I believe this is discussed slightly more later in the paper, but is omitted here.
- 9) Line 57: Should there be punctuation after 'on the unit sphere'?
- 10) Line 98: Sigmoid shouldn't be capitalised here as far as I'm aware.
- 11) Line 143/144: Appears to be a strange line break in the notation here.
- 12) I think there needs to be more clarity on the usage of 'dataset' through the paper. As I understand for INR, one subject = one dataset, but this may be confusing to readers given a dataset often refers to a set of subjects for many machine learning tasks.
- 13) In the future work section, it could be worth including the possibility of extending the work to a multi-dataset (i.e. multi-subject) setup, where the model parameters are shared across subjects (see Wang et al., 2024. SCARF: Scalable Continual Learning Framework for Memory-efficient Multiple Neural Radiance Fields, for instance). This seems to be a natural next step to get sublinear increases in train time and model size as the number of subjects increases.

Reviewer #3

(Remarks to the Author)

I thank the authors for addressing my comments satisfactorily, in particular, their comprehensive evaluation of the Ground Truth generation (removing the smoothing step and exploring different sigma maps and protocols), and although they did not perform the comparison I suggested, I am satisfied with the comparison against NLLS fitting with varying protocol per voxel and the discussion about the non-trivial limitation of their gradient non-uniformity correction approach.

I do not have further comments

Version 2:

Reviewer comments:

Reviewer #1

(Remarks to the Author)

Authors have appropriately addressed the questions raised in the previous review. I thank them for their effort and work. I am only pointing to an error that persists across versions in equations 2 and 3.

Minor comments

1. In Eq.2, the parallel and perpendicular components of the second operand are swapped (they have been swapped since the first version of the manuscript) when comparing to Tax et al. 2021, where it is written as $\frac{1}{3}b*(D_{\text{perpendicular}} + 2*D_{\text{parallel}})$. Is there a reason for the swap? If this is an oversight, when fixing it, please note that Eq. 3 will also need to be changed accordingly.

Reviewer #2

(Remarks to the Author)

Summary

This paper presents a deep learning-based method for signal estimation, grounded in the Standard Model of white matter. Their work is inspired by the effectiveness of implicit neural representations (INR)--a more general deep learning technique for representing a signal, e.g. an image, as a continuous function(i.e. rather than discrete pixels). They evaluate their method across both synthetic and in-vivo datasets. They compare their approach to 3 other methods and find their INR approach delivers broadly superior results.

Comments

I would like to thank the authors for the additional changes and feedback following the second round of reviews. I believe all of my concerns raised and clarifications requested have now adequately addressed by the authors.

I have no further comments.

1. General comments to the reviewers

We thank the reviewers for their valuable input, and we have made significant efforts to incorporate the requested additional information into the manuscript. The main additions include a comparison of the effects of gradient non-uniformity correction relative to NLLS (see results, section 3.5 and Fig.R4), a detailed description of ground truth generation along with the impact of alternative methods for generating the ground truth (Fig.R6-R7), simulations using an acquisition protocol not optimized for SM fitting (see supplementary information section 4, Fig.R5), visualization of FODs for the in vivo dataset across different l_{max} values (see Fig.R3), and improved anatomical alignment in Fig.R2. We believe these additions significantly strengthen the manuscript and provide a clearer, more comprehensive presentation of our findings.

2. Reviewer 1

Authors propose to use implicit neural representations (INRs) to compute the standard model (SM) parameters for white matter tissue employing diffusion MRI (dMRI) data. The present work builds upon previous work by the authors (Hendriks et al. *Imaging Neuroscience*, Volume 3, 2025), where the exact same approach and neural network architecture (up to the output heads) is used to compute the spherical harmonic coefficients on white matter tissue and volume fractions for gray matter and cerebrospinal fluid. In the present work, the input data positional coordinates are mapped to some higher frequency domain through Fourier encoding to finally train a neural network to estimate the SM parameters. The training is fully unsupervised, as the SM parameters are fed to a forward model to compute the dMRI signal which drives the optimization. A number of experiments are presented on simulated and in vivo data in order to study the impact of their method. These include varying the noise level, using Rician noise, or simulating gradient inhomogeneities. Compared to conventional or other supervised machine learning and neural network approaches, authors show that their method outperforms them in terms of the error and correlation with respect to the known ground truth. Graphic results provide a qualitative sense of the variance and bias of each particular method. Authors claim that their method could be used to provide SM estimates that are robust to noise, gradient inhomogeneities, and within reasonable inference time.

The experimental design is limited to the case of a single brain dMRI dataset, including simulated and in vivo data. Although the evidence presented in terms of results shows that INRs improves the compared methods, overall, the training process of the neural network

remains unclear, and the generalization to $N > 1$ and other acquisition settings remains to be demonstrated.

Thus, although the body of work presented holds promise, a number of concerns and questions exist that would need to be addressed.

R1.C1 Authors mention the long/extensive acquisitions in the abstract as a challenge for computing the SM parameters, but do not later (e.g. in the introduction) develop why long acquisitions are not ideal (even if evident). They also mention "INRs demonstrate potential to improve on voxel-based methods to estimate parameters, especially in (...) sparsely sampled datasets" (line 32). However, both datasets used in the experiments are issued by such extensive acquisitions. Thus, it is unclear how the method applies to data that has not been acquired (or simulated) using a dense sampling of b-values, directions and b_δ values. In order to support the claim that an INR does not require training data from extensive acquisitions, it would be necessary to create a subsampled dataset (directions, shells, and b_δ) and show that training on subsampled data provides accurate estimates wrt those on full data.

Response

Thank you for this comment. The abstract of our original manuscript admittedly seemed to imply that we want to use INRs to fit the standard model to reduced dMRI datasets. Instead, we showcase performance on a previously published 'optimized' protocol of ~15 minutes to generate a synthetic dataset, which is a time optimized protocol. For the purpose of fitting the standard model, this protocol is already sampled rather sparsely. We have adapted the abstract to better reflect our intentions, i.e. that the purpose of the current manuscript was not to find optimized reduced acquisition protocols. We have adapted the mentioned sentence in the abstract to: *However, due to the model's high-dimensional nature, accurately estimating its parameters poses a complex problem and remains an active field of research, in which different (machine learning) strategies have been proposed.*

Related to this, also addressing R3.C2, we have investigated how the methods behave when applied to another acquisition protocol, i.e. of the in vivo dataset used in this study, which does not contain high b-values ($b \geq 6000$). Although not a further reduced protocol, these experiments show generalizability to other acquisition protocols. These results have been added to the manuscript (see supplementary information section 4). The INR method shows only a very slight decrease in performance under this non-optimal protocol, while the other methods exhibit a more significant drop. This experiment was performed at SNR 50.

We agree that a more systematic investigation of the response of the INR method to sparser, clinically feasible acquisition protocols would be highly interesting and represents a valuable direction for future work. However, such an analysis would require a substantial amount of additional research (see, for instance, (Álvaro Planchuelo-Gómez et al., 2024), which is fully devoted to sparse sampling of acquisition protocols) and lies beyond the scope of the present study, as protocol optimization was not the original aim of our work.

R1.C2 It is unclear from the introduction what the novelty or originality of this work is with respect to previous scientific literature that have employed INRs to model dMRI data, including authors' previous work (Imaging Neuroscience, vol.3 2025). In that work, authors used the same exact approach (network architecture, Fourier encoding, unsupervised training) to compute SH coefficients (which are also computed in the present manuscript, besides the SM parameters). So readers would benefit from a clearer motivation of the work and distinction with respect to previous work.

Response

We agree with the reviewer that the design of the framework for this work and for the previous work has similarities. However, the goal of this work is to use INRs as a way to progress the open area of research concerning the fitting Standard Model parameters. Extending the model from previous work to also include kernel parameter estimation is a significantly harder problem to solve and provides many potential advantages, which are mentioned at the end of the introduction (see also "Strengths" by reviewer 2, and general comments by reviewer 3). Furthermore, the analysis we have performed and the comparisons to existing machine learning methods provide more insight into the performance of the INR framework, and provide a better way to position this methodology in the current literature. We have adapted the introduction to more clearly state this (line 36): *Building upon our previous work ((Hendriks et al., 2023); (Hendriks et al., 2025)), we extend INRs to estimate the SM parameters alongside the FODs*

R1.C3 Authors mention that "(The SM) issues become even more prominent at high noise levels" (line 15) and "INRs demonstrate potential to improve on voxel-based methods to estimate parameters, especially in more noisy (...) datasets" (line 32). However, in practice, results for a single noisy setting (SNR 20, SNR 50 being closer to the noiseless side) are presented. A larger number of low SNR values would be required to demonstrate that INRs provide consistent improvements on low SNR values.

Response

The SNR described in this paper is the SNR on the unweighted (b_0) image. The SNR on the diffusion weighted images decreases as the b-value increases (with the attenuation of the signal). For diffusion MRI the SNR values of the unweighted signal typically lie between 25-100, thus with SNR 20 we are on the lower side, and 50 can be considered representative SNR for dMRI acquisitions, especially with developments in hardware performance that are becoming more commonplace. We have adapted the method section to state this more clearly (see lines 157,158)

R1.C4 The sentence "It also does not require uniform acquisition parameters across voxels, making it capable of explicitly correcting for gradient non-uniformities (...)" is misleading. Unless the shimming completely removes these, gradient inhomogeneities are likely present on any acquisition, and more prominent at stronger field strengths. These can be minimized when processing the diffusion data. Also, these variations are not deliberate; however, the first part of the sentence suggests that INR can perform well on settings where each voxel is deliberately acquired with different acquisition parameters (e.g. b-values, b_{delta}). Given that acquiring each voxel with different parameters has already been proposed in the literature, in order to support the claim that INRs are able to model data purposely acquired with different parameters, an experiment that demonstrates it would be needed.

Response

We agree with the reviewer that the original sentence could be misinterpreted and may cause confusion. Our intention was to clarify that, although a uniform acquisition protocol is chosen during scanning, gradient non-uniformities alter this protocol depending on the voxel's position in the scanner, thereby generating a spatially varying acquisition protocol that is indeed not deliberate. Here, we merely aimed to show that spatially-varying B-matrices due to gradient non-uniformities can be taken into consideration within the INR framework (which is much more cumbersome in supervised learning) and we are not considering the deliberate use of different acquisition parameters per voxel; therefore we have not included such an experiment. We have clarified this statement in the introduction of the manuscript as follows (lines 45-46): *Furthermore, it is capable of explicitly correcting for gradient non-uniformities by adapting the acquisition protocol per voxel with the spatially varying gradient coil tensor of the scanner (Bammer et al., 2003).*

R1.C5 It is unclear how the training/testing is performed for the INR model. What is the training and testing data used? How is the data split into training and testing? Since $N=1$ in the work, it is unclear how the INR was trained.

Response

We kindly refer the Reviewer to our next comment (R1.C6).

R1.C6 In section 4.1 it is mentioned that "the supervised NN method (...) can be significantly more time-consuming due to its reliance on NLLS for generating part of the training data and the need to retrain the model for each specific acquisition protocol.". However, authors do not demonstrate the performance of INR when applied to another acquisition protocol, so a new experiment needs to be added to demonstrate how well their method performs compared to the supervised NN when being applied to another acquisition protocol on which INR has not been trained. This is related to the passage "INRs require a model to be fit to every individual dataset." (line 270). When using a neural network, the expectation is for it to be trained once on some data and then be applied many times to different data. So it is unclear what the gain of using an INR is in this context. Also, if it has to be retrained on every individual dataset, it is unclear how the learning process (training/testing splits, etc.) would proceed.

Response

Indeed, the proposition of INRs is somewhat different from 'regular' machine learning methods, which understandably can lead to some confusion. INRs use a neural network to create a representation for a single dataset (e.g., in a self-, or unsupervised way) and, therefore, require a single network to be fit for every dataset one wants to represent. There is no train/test split - as done with Supervised Learning. The estimated parameters arise as output of the MLP, and the quality of the fit is evaluated through different methods (both quantitatively using the synthetic ground truth, and qualitatively on real data in our case). This has advantages and disadvantages. Major advantages include: leveraging dataset specific spatial correlations, not needing a training set (thus, no training set bias or work intensive labeling), flexibility in acquisition schemes (where supervised methods are trained for specific acquisition schemes), possibility to correct gradient non-linearities (see R1.C4), amongst others (see discussion). The main disadvantage is what the Reviewer mentions in comments C5 and C6: the need to fit an INR for every individual dataset. Although this training process is much quicker than fitting with NNLS or training a supervised network (minutes, see Table 3), the time can add up for databases with many subjects. Optimizations to more quickly fit INRs, such as hash-encodings (Dwedari et al., 2024), exist but are outside the scope of this paper. To assess the robustness of our model to acquisition settings we fit the model to two different datasets (the synthetic and the in vivo dataset), which are acquired using different protocols. We have changed the wording in our method section to make the general idea more clear (lines 79-80).

R1.C7 In Table D.4 (supplementary information section 4) results for synthetic noiseless and SNR 50 data are shown, and it is argued that INR is robust to noise when computing parameter maps for different l_{max} values. However, SNR 50 is arguably not a noisy setting. Results with SNR 20 (and potentially other low signal values) would be required to demonstrate this.

Response

We agree with the reviewer that it is interesting to show the SNR 20 results as well. We have updated Figure R2 to now include the SNR 20 synthetic dataset and have moved the in-vivo dataset to a separate figure (Figure R3).

R1.C8 Figure 1 shows the output values computed by INR, including the diffusivities, the compartment fractions, S_0 and S_H coefficients. However, it is mentioned that (section 2.5) the p_2 rotational invariant will be used to demonstrate model performance, and such is the

case across the figures in the manuscript. However, tables D.4 and D.5 employ S_0 instead of p_2 . What is the reason for such change?

Response

Thank you for pointing this out. The main rationale behind this was that the figures in the main text show that there is not a large difference between the FODs (which is quantified in part by p_2). It is indeed interesting to see the p_2 value average errors. We have added these values, and the other parameter values for SNR 20 to D.4 and D.5.

R1.C9 When authors mention the inference time (section 4.1, line 269), they provide an example with l_{max} 2: although l_{max} 2 is heavily used across their experiments due to limitations of the models they compare to, it would be informative to measure the time for other l_{max} values enabled by common modern acquisition settings (e.g. more than 15 directions at least, and maybe 30, even on a clinical setting). What is the inference time for e.g. l_{max} 4, 6, or 8? Is the increase linear as the number of SH coefficients that need to be estimated increases?

Response

We kindly to point the reviewer to Table 3, which shows the fitting times for all models we have fit for this manuscript. We indeed note an increase in fitting time with increasing l_{max} that appears to be more or less linear with the number of parameters being fit.

R1.C10 In section 4.1 authors mention "However, since INRs require a model to be fit to every individual dataset, supervised inference remains faster for datasets consisting of many acquisition parameter settings". What do authors mean by "every individual dataset"? What is the meaning of "dataset" in this context (i.e. does this mean data composed of multiple participants, or data containing multiple acquisition parameters)? Given that $N=1$ in the manuscript, How does the authors' method work when there is multiple subjects? This question is related to the previous questions about the training/testing splits of INR (questions 5, 6).

Response

Please see our answer to comment 6. We have changed the wording in this particular sentence to clarify the meaning of 'dataset' (see lines 280-283): *However, since INRs require a model to be fit to every individual subject, a supervised learning approach could remain faster for datasets consisting of many subjects with identical acquisition protocols, despite the considerable amount of training time it requires initially (e.g. 83 minutes on GPU excluding initial NLLS parameter estimations, for the supervised NN method (de Almeida Martins et al., 2021))*

R1.C11 The abstract contains some missing punctuation marks/whitespaces after punctuation marks or excess of these. Please fix those, e.g. "(...) existing methodsResults demonstrate (...)" "(...) plausibly in a continuous way.The INR is (...)" "(...) achieves fast inference , is robust to (...)" "(...) corrections.The combination (...)"

Response

Thank you for pointing this out, we have fixed these issues.

R1.C12 Line 6: Why is "B" in "B-tensor" capitalized? The rest of the diffusion sensitization weighing notations use the common, lowercase "b". Applies to line 149 as well.

Response

When talking about b-values and b-vectors, indeed the lowercase applies. Here, the B tensor is a matrix that describes the strength, orientation, and shape of the diffusion encoding (linear, planar, spherical). The common convention is to use a capital letter to describe this matrix (Westin et al., 2014), and we have adopted that in our manuscript.

R1.C13 Line 8: "Alzheimer disease": "Alzheimer's disease".

Response

Thank you, we have fixed this typo.

R1.C14 Line 52: "standard model" is not capitalized in here whereas it was in previous mentions to it. Adopting a single case would be more consistent.

Response

We have updated all occurrences of "Standard Model" to now be capitalized.

R1.C15 Line 56. Specify what "n" (bold n) is in equation (1), even if it is evident.

Response

We have carefully checked whether n is specified (line 54 of our original submission, above equation 1).

R1.C16 The notation around line 61 is misleading: why do authors assign (from the notation) the entire diffusivity of the extra-axonal compartment to the parallel component (i.e. $D_{parallel} = D_e$), even if they model the extra-axonal compartment as a symmetric tensor having both parallel and perpendicular components? Upper/lowerscripts should be used at least to avoid making the notation confusing.

Response

We thank the reviewer for this comment. In our notation, the extra-axonal compartment is modeled as a zeppelin with axial and perpendicular diffusivities, $D_{\parallel} = D_e$ and $D_{\perp} = D_p$, respectively, where D_e specifically denotes the axial diffusivity rather than the total extra-axonal diffusivity. We hope this clarifies the intended meaning.

R1.C17 A revision of equations 2 and 3 is necessary: equation (2) has a missing closing bracket in the second operand, which seems to have been mistakenly placed at the end. In equation 3, the first operand is missing the term b , and probably the factor.

Response

We agree with the proposed modification of the reviewer and implemented it in the manuscript.

R1.C18 Please, use a single line for equation (2) to make it easier to read.

Response

We agree with the proposed modification of the reviewer and implemented it in the manuscript.

R1.C19 Please, use the same term to refer to the first block of the network: "spatial encoding" vs. "positional encoding". This applies to the appearances in the text and the name of section 2.2.1.

Response

We have updated Figure 1 to now state 'spatial encoding' in the first block, rather than 'positional encoding'.

R1.C20 What do authors mean by "signal + coordinates" in the Figure 1 caption? According to the text and the operation of an INR, only the coordinates are used as the input data. If "signal" refers to the additional data used at the input in Experiment 5, please make it explicit.

Response

Thank you for pointing this out. The word "signal" was mistakenly added in the description and has been removed.

R1.C21 Add the min/max values corresponding to the SH coefficients to Table 1, even if it is to say that they can be $-\infty$, $+\infty$ (or -1 , $+1$).

Response

Table 1 has been updated to include the SH coefficients.

R1.C22 Authors mention in the Implementation section (line 113) that they used "default settings otherwise": what do "default" settings mean in this context? What are those parameters?

Response

We have included the settings in the description of the optimizer (line 116).

R1.C23 Please, specify what l_{max} was used in "Experiment 2: Rician noise bias".

Response

We have added a sentence to specify this (line 178).

R1.C24 How where the SM values computed for the Cardiff-acquired data to drive the optimization of the neural network?

Response

We kindly refer the reviewer to our explanation about the fitting process of INRs under C6. It is not required to have the values of the Standard Model parameters to fit the model, as this is not a supervised learning approach.

R1.C25 In section 2.8.5 authors mention that their neural network receives additional parameters to model the gradient inhomogeneities. These parameters should probably be introduced in the methods, as this means that the neural network not only receives the x,y,z vector as the input, but also additional parameters.

Response

Thank you for pointing this out. We appreciate that from our way of wording it previously might have seemed like additional inputs to the neural network. This is not the case however: the spatiotemporally corrected B-matrices (which are slightly different for each voxel, instead of having one b-vector and b-value for the entire volume) are input to the forward equation for signal prediction. Note that the B-matrix is input at this level for all experiments, but now it is corrected for gradient non-uniformities. We have changed the description of the experiment to more clearly state this (line 195).

R1.C26 The resolution of the simulated data is not specified: besides being informative, it seems relevant for "Experiment 6: Implicit neural representation for spatial interpolation". Readers can only do the reverse computation, but it would be good to save this task to them.

Response

Thank you for this comment. We have added the original resolution to the description of the dataset (lines 142, 155-156).

R1.C27 How do authors explain the effect that INR exhibits some overestimation of D_e in the splenium (section 3.1) or again the differences in the splenium and other structures in section 3.3?

Response

Our hypothesis is that it has to do with the generation of the ground truth dataset. In the splenium area the parameter maps produced by SMI during the generation process are not that spatially correlated, which is especially apparent in the D_p . As parameters are estimated jointly, the incorrect estimation of the D_p might lead to incorrect estimation of the D_e . We see similar behavior in the supervised NN. SMI might deal with this better because it was used to generate the ground truth and therefore is able to produce this pattern of parameters. The differences in the in-vivo dataset are probably caused by the spatial correlation that is present in the INR output, as the differences are much more spread out across the brain and smaller than in the synthetic dataset.

R1.C28 How do authors explain that when no noise is added, INR does not perform as well (section 3.1)?

Response

The INR does perform better when there is no noise added, compared to itself in noisy scenarios. The lower performance in the noiseless case compared to other methods can be explained by the spatial correlations in the INR output. Since it does not model each voxel individually and the synthetic ground truth is not perfectly spatially correlated, it is unable to capture the input data perfectly which will cause some bias. The noiseless experiments were included to study the extent of this bias. The INR could get closer to the ground truth in noiseless scenarios by increasing σ and the MLP size. This is not desirably in noisy scenarios, however, as it would overfit on the noise. Hence, there is a trade-off between spatial regularization, noise-robustness, and introducing bias. We added the following sentence to the discussion (lines 298-300): *The lower performance of the INR method on simulated data without noise arises from its inability to capture the input data perfectly, as it does not model individual voxels, unlike the other methods.*

R1.C29 The Figure 7 caption says "Difference maps are with respect to prediction with MSE and without gradient non-uniformity correction (top row)". Although the reader can

assume that the "Corrected+MSE" was computed wrt the "uncorrected" version and the "Corrected+Rician" wrt to the "Corrected+MSE", please make this explicit; if the computations were not these, please be more clear in the caption.

Response

The corrected versions were both compared with the uncorrected versions. We adapted the figure caption to make sure this is clear for the reader. Please note that this figure is adapted as the Corrected+Rician is moved to the supplementary information section 5 and is replaced by NLLS with gradient non-uniformity correction. For more info on this see comment R3.C7. (See Fig. R4)

R1.C30 In section 4.3 authors mention "When sampling outside of the white matter (as shown around the edges in Figure 8), the INR produces inaccurate results.". It is unclear what authors refer to as discrepancies are not apparent in the figure. If there are, these should be mentioned in the results and pointed with an arrow to then discuss the issues/causes/hypotheses/mitigation measures in section 4.3.

Response

We agree that these discrepancies were not apparent and have removed the reference to the figure entirely. The main point of the primary sentence was to state that it is only applicable to white matter and that extension to other models could be useful.

R1.C31 Please, revise the missing letters in author names and title in reference 55.

Response

Thank you for the detailed examination of the reference list. We have fixed this issue.

R1.C32 Section 4.1 and supplementary information section 2. Do not use "Method 1", "method 2", "Method 3", etc. Please, name the methods as you have done throughout the rest of the article.

Response

Thank you for pointing this out, we have changed the names to be more descriptive and match the rest of the manuscript.

R1.C33 Appendix B. Line 536: "(...) where Gaussian noise is considered with SNR 20". It should be "SNR 50" according to the text in section 3.1, the caption and the fit of the values shown in the figure.

Response

We have fixed this typo.

R1.C34 Please, adopt a consistent naming for the "noiseless" data: noiseless: 7 apps; clean: 2 apps. Sticking to one would make the read easier.

Response

We agree that consistent terminology is preferred and have chosen to adopt 'noiseless'.

R1.C35 Lines 249 and 252: "The quantitative experiments on synthetic data show how the proposed method outperforms existing methods on noisy data." and "At higher noise levels (SNR = 20), the INR method clearly outperforms all other methods (see Fig. 3)". These sentences convey the same message/one is a reworded version of the other, with a passage about less noisy data in between. Please, reformat the passage so as to avoid the fragmentation.

Response

Thank you for pointing this out. Line 249 of the original manuscript has been removed to avoid duplicate statements.

R1.C36 The parameter sorting across tables and figures (columns) is not consistent: Table 1, figures and Tables D.4 and D5 use all a different sorting for the columns (figures are consistent among themselves, and table D.4 and D.5 are consistent between themselves). Please, sort the columns so that the same sorting is used across all tables and figures.

Response

Thank you, we have adapted Table 1 and the tables in supplementary information section 4 to now reflect the same order of parameters (with inclusion of S_0) of the Figures and the text.

R1.C37 In section 4.1 authors mention "However, since INRs require a model to be fit to every individual dataset, supervised inference remains faster for datasets consisting of many acquisition parameter settings". The term "supervised inference" is uncommon and unclear. Do authors mean "supervised training"?

Response

We indeed meant supervised learning, we have adapted the text to reflect this. Also see comment 10.

R1.C38 Line 286: The sentence "(...), we have attempted to reduce biases towards the estimation for creating the ground truth by smoothing the parameter maps and (...)" is difficult to read/is not well understood (especially, the part "towards the estimation for creating the ground truth"). Please, re-word.

Response

We agree that this is a confusing sentence and have changed the wording to be more clear.

R1.C39 In section 4.2 authors state "Additionally, correcting for gradient non-uniformities has a significant impact on the parameter estimates, yet the accuracy remains to be evaluated. Both experiments in section 3.2 and 3.5 show (...)". The results in section 3.2 are not about gradient non-uniformities. Is this a typo?

Response

Thank you for pointing this out. This is an error related to how the experiments were structured previously, and has been corrected.

3. Reviewer 2

Summary

This paper presents a deep learning-based method for signal estimation, grounded in the Standard Model of white matter. Their work is inspired by the effectiveness of implicit neural representations (INR)—a more general deep learning technique for representing a signal, e.g. an image, as a continuous function (i.e. rather than discrete pixels). They evaluate their method across both synthetic and in-vivo datasets. They compare their approach to 3 other methods and find their INR approach delivers broadly superior results.

Strengths

-S1 Their approach requires only the signal to reconstruct at a particular location. The SM parameters are learned implicitly as a sort of interpretable latent variable. Due to the long latency of baseline methods, this approach is generally much faster including train + inference time. -S2 INR, unlike voxel-based approaches, is resolution agnostic. Theoretically, the model could make predictions at inter-voxel locations, enhancing its downstream use-cases. -S3 The proposed method outperforms all three baselines, particularly at higher noise regimes relative to supervised approaches, suggesting their approach may be more robust. -S4 The method is highly extensible to learning any arbitrary parameters provided their relationship with the signal can be modelled in a differentiable way. -S5 The code is made publicly available, facilitating reproducibility of the results.

Weaknesses

-W1 As the claims of the paper are not only that INRs are more efficient, but also more accurate than supervised methods, it would be useful to see an ablation on where these

performance gains are coming from. Specifically, applying some additional supervision via an auxiliary loss function to predicting the intermediate SM parameters. As the supervised NN baseline does not use an INR framework, this ablation would provide a better apples-to-apples comparison.

Response

We thank the reviewer for this insightful suggestion. We agree that exploring this could provide a more direct comparison to supervised baselines. Previous work has explored this in autoencoder networks where the input was the signal and the loss function was defined on the prediction of the model parameters, but found suboptimal performance compared to predicting the signal through a forward equation (Álvaro Planchuelo-Gómez et al., 2024) . Several possible issues arise when integrating such levels of supervision, for example how to weigh parameters with different ranges, how to appropriately integrate differences in parameter precision in the loss (Zhang et al., 2022), and the fact that a ground truth is needed. In the case of INRs, which are fit to each dataset separately, such ground truths (even based on simulations) are impractical to include. Hence, we suggest this analysis to be beyond the scope of the current work, as our focus is on demonstrating the advantages of INRs in their native self-supervised formulation.

Regarding the origin of the performance gains of the INR, we have conducted experiments by varying characteristics of the ground truth to see whether the superior performance of the INR method is generalizable. We included simulations where noise was varied (see supplementary information section 2). This analysis showed that at lower noise levels (SNR 50), the performance gap between INR and the other methods decreased, whereas in the absence of noise, INR performed worse compared to the alternatives. This suggests that the overall performance advantage of INR primarily stems from its superior ability to exploit structural coherence – which is expected in tissue anatomy. Furthermore, we also included results on simulations where the smoothing filter of the ground truth was omitted following request of reviewer 3 (see R3.C1). The results are shown in Fig. R8. The results indicate that without the smoothing filter, INR performance declines, although the other methods are also affected.

-W2 The terminology around 'unsupervised' could be refined. INR is not generally considered to be an unsupervised technique as the signal is a label. I understand it could be considered 'unsupervised' with respect to the SM parameters, but the terminology could be confusing as the proposed technique is closer to indirect/weak supervision.

Response

We agree with the reviewer that referring to the method as “unsupervised” is not entirely accurate. If acceptable to the reviewer, we propose adapting the terminology to “self-supervised,” as the network is guided by a loss function directly connected to the original input. The manuscript has been revised accordingly.

Other Comments & Typos

42-43 'Moreover, because this is an unsupervised method, it does not rely on large training datasets as supervised methods do'. This claim needs support—generally supervised methods are more sample efficient than unsupervised ones in deep learning.

Response

We agree that this was unclear and have rewritten the sentence to be more descriptive of the process.

Abstract '...way.The INR ...' there appears to be no space before 'The', unless its just a latex artifact.

Response

Thank you for pointing this out. There were multiple issues with unwanted and missing whitespaces in the abstract, which have now been fixed.

4. Reviewer 3

In the present work, the authors propose the use of implicit neural representations (INRs) to estimate the parameters for the Standard Model (SM) of diffusion in white matter (WM) from diffusion MRI data. Authors have successfully used INRs previously to solve the fiber orientation distributions only, but now they extended their technique to estimate the full set of SM parameters. They argue that the proposed approach improves over existing SM parameter estimation methods as it uses spatial regularization, has the ability to explicitly correct gradient non-linearities, and avoids biases from training data due to its unsupervised nature. They compare their method with two state of the art techniques: standard model imaging (SMI), and a supervised deep learning method (Supervised NN), showing better results, specially at lower SNR.

The contributions in the method proposed by the authors: spatial regularization, correction of gradient non-linearities and non-biases because of training data, are important and necessary upgrades for SM parameter estimation, and the authors did a good job describing their

approach and its advantages in the article. I was impressed on how good their results are for the synthetic data evaluation, in comparison with the other methods. However, I still have a few questions/comments I'd ask the authors to respond.

R3.C1 I have some doubts regarding the procedure for the simulated ground truth (GT) data creation. The MGH Connectome Diffusion Microstructure Dataset consist of scans from 26 subjects, in which 7 of them were rescanned, is there any strong rationale for using the scan from only one subject (not even rescanned) for the GT creation? Why not using the scans/rescans from all the subjects? It's my understanding that although these scans were preprocessed, they were not denoised. Then, SNR can be highly increased by using the average from all subjects (either by averaging the dwi's or averaging the resulting parameters, after warping them into a common space), then, it would be probably fine to use NLLS (with multiple initializations), for the computation of the GT parameters, which probably will be less biased than the strange combination of SMI parameters plus MSMT-CSD FOD to generate the synthetic signals. Also, spatial coherence will be enhanced naturally on the averaged cohort instead of artificially applying smoothing on the kernel maps, which may give unfair advantage to the INRs method.

Response

Many thanks for the suggestions. We acknowledge that the creation of a ground truth (GT) is inherently a subjective process that involves many choices, and that our method is just one way of realizing this. To our knowledge, there is no established standard procedure for GT generation in this context. In our work, we experimented with several approaches and ultimately chose the one that, in our view, provided a sufficiently realistic balance between anatomical reality and controlled assessment. Our selection was guided by two main criteria: (i) obtaining a plausible distribution of Standard Model parameters, and (ii) ensuring structural coherence, which is to be expected in brain tissue and is a key reason why INRs are particularly suitable for microstructure fitting.

We fully agree with the reviewer that alternative strategies could be employed, and some may indeed improve certain aspects of GT generation. We therefore investigated whether changing the GT generation per the various suggestions of the reviewer would drastically change parameter distributions (see detailed answers in the next responses). As overall conclusion, and to maintain consistency and comparability throughout the manuscript, we would prefer to keep the current GT definition largely unchanged if the reviewer agrees. To answer the concerns of the reviewer and support the GT validity, we have included additional experiments (see Fig.R6-R8). Specifically, we show results on simulations where the smoothing step of the ground truth is omitted, in order to assess the extent to which INRs benefit from the imposed spatial coherence. A performance decrease for the INR method is observed, as expected, although the other methods also exhibit a decline. In conclusion, without the smoothing filter, the INR method still outperforms the other methods. This supports the hypothesis that the main performance gain of INR arises from its superior handling of noise. This effect is further illustrated by the simulations in supplementary information section 2, which show reduced INR performance when fitting is performed on simulated signals without noise, where a bias can be expected due to spatial regularization.

Regarding the suggestion to average across multiple subjects: while we appreciate the potential SNR gains, averaging across different brains could blur or distort individual anatomical structures, thereby reducing the biological plausibility of the resulting dataset.

R3.C2 Also, while using an optimized protocol for the simulated GT data was great, it could also be good to report how the methods behave with the protocol of the real in vivo data used in this study, as it does not contain any high b-value ($b \geq 6000$). This could evaluate the biases of the methods due to a not optimal protocol.

Response

This is indeed an interesting avenue, and we have now added these experiments to the manuscript (see supplementary information section 4, Fig.R5). The INR method shows a very slight decrease in performance, whereas the other methods show a more significant decrease in performance. This experiment was performed with SNR 50.

R3.C3 Finally, SM is valid for long times, the MGH dataset contains data from two diffusion times, $t=19$ and $t=49$, authors used $t=19$, but why not use the data with the longest time? This could be more appropriate and less biased for SM, and contains enough data for the parameter estimation, even if $b=200$ is excluded (because it contains considerable free water signal, which is not considered in the fitted model) and even maybe excluding $b>10000$ (if SNR is too low).

Response

We agree with the reviewer that this could introduce a potential bias. We investigated differences arising from using datasets with different diffusion times as the basis for the ground truth, as well as varying sigma maps as input for the SMI fitting (as suggested in R3.C4). We have added Fig. R6-R7) describing the effects of these choices on the parameter distribution of the ground truth. Our interpretation is that the estimates from different diffusion times are correlated, the differences between the original ground truth and the alternative fits are acceptable and may be attributed to the degeneracy of SM fitting under the LTE protocol. The overall parameter distributions remain very similar. Therefore, we propose to retain the current ground truth, which also maintains consistency within the manuscript.

R3.C4 I also have questions regarding the use of the MPPCA method to compute the noise map. As the authors mentioned SMI needs a noise map, which is used to regularize the estimation according to the SNR of the fitted signals (if I understand correctly, although it does not regularize spatially, its function could be somehow analog to the function of hyperparameter σ of the INRs method in the sense of how smooth the resulting parameters maps could be). Well, as mentioned in the previous point, the dataset used for the GT creation was preprocessed but not denoised, then using MPPCA to estimate the noise map it's probably not correct as there are correlations in the noise of the data which violates the assumptions of the MPPCA method. Then, MPPCA could very likely underestimate the noise map. If you compute the data SNR by dividing (b0 map)/(noise map), which is this value in WM on average? According to the owners of the data, it should be around SNR=23. If this value, is much higher, then SMI parameter maps may be less smooth than they could be. Hence, this could be another reason to change the approach for the creation of the GT as I mentioned in the previous point. Also, if using this MPPCA noise map to correct the Rician bias, then the bias not be fully corrected. It may be better to compute sigma from the 50 b0s in each scan, or use the 14 subjects (7 with rescan) with real-valued data in the MGH dataset.

Response

We agree with the observations of the reviewer. Regarding SNR values, we performed calculations and found $SNR \approx 24$ in white matter when using the standard deviation of the 50 b0s. This corresponds to the value reported in the article of the HCP data. MPPCA on b0s finds $SNR \approx 36$ and MPPCA on all dwi volumes $SNR \approx 50$. This finding supports the reviewer's concern that applying MPPCA to all DWI volumes may underestimate the noise, potentially leading to less smooth SMI fits.

Together with the previous comment we performed experiments with different methods to fit SMI and compared it to the SMI fit which was used for generation of the ground truth. See comment above for our interpretation of the results presented in Fig. R6-R7.

Due to this discussion, we also reconsidered the SMI fitting on the in vivo data. The fitting was redone with a MPPCA map calculated on the unprocessed data with Rician noise correction. The INR was subsequently also redone with Rician noise correction. The figure and text were adapted accordingly (see Figure R3 and lines 183-184).

To sum up these discussions on ground truth generation, we added the following to section 4.2 (Limitations) of the manuscript (see lines 308-316): *Omitting the smoothing still resulted in the highest performance for INR, see supplementary information figure 7. Another limitation related to the ground truth is that the MGH dataset used in this study contained only LTE, which may have led to inaccurate parameter estimates (see discussion at the end of this section). We have investigated the impact of other possible sources of severe bias on creating ground truth parameter maps from the MGH dataset, such as the relatively short diffusion time, noise estimation procedure, and included b-values (results not shown). We found similar overall distributions and linear voxel-wise correlations when using longer diffusion times, noise estimate from repeated $b=0$ s/mm² images, and excluding $b=200$ s/mm² and $b>10.000$ s/mm² images. Nevertheless, creating a ground truth that balances capturing anatomical reality while exerting sufficient control remains an important avenue to further explore.*

R3.C5 Regarding the estimation using non-linear least squares (NLLS), when working on noisy data, its good practice to use multiple initializations (and then keeping the best result), as the degeneracy of the SM estimation problem, in which there are multiple optima, is well known. Was this multiple initialization approach used for NLLS estimation? The manuscript mentions that the Levenberg-Marquardt algorithm was used with max 1000 iterations, but it does not seem this refers to 1000 initializations, is it?

Response

We apologize for the lack of clarity in our original description of the NLLS estimation procedure. The reported 1000 iterations refer to the maximum number of iterative steps allowed for the Levenberg–Marquardt minimization, and not to the number of initializations. In practice, we performed the fitting with two different initializations, and retained the best result. While we considered using a larger number of initializations to further mitigate the risk of local minima, we did not perform this as the fitting time for NLLS would greatly exceed other methods and would thus not be a fair comparison. We have revised the Methods section of the manuscript to explicitly describe this aspect of the NLLS fitting procedure (section 2.5).

R3.C6 For the evaluation of the upsampling capabilities of the proposed method, 8x of the original resolution, authors compare with simple 3d interpolation methods, which although useful for simple tasks, are probably not be the best to perform the comparison. Why not compare against more appropriate upsampling methods from literature? I can think for example on the Non-local MRI upsampling (Manjon et al. Medical Image Analysis. 2010), with codes available in the webpage of the first author. Although, this method could be too old already, maybe there are newer available method for upsampling.

Response

We agree that we are not comparing to the state-of-the-art upsampling methods. Showing the upsampling capabilities of the INR has been presented in (Hendriks et al., 2025) for fODFs and was not the primary focus of this manuscript; i.e. the fitting of SM parameter estimation. However, to show that the INR creates a spatially consistent representation -and is not just reproducing values at the exact coordinates- we have added this experiment. Exploring this further is very interesting and in our opinion is a sufficiently complex area that it warrants a separate publication.

R3.C7 Similarly, for the evaluation of the method capabilities to correct gradient non-uniformities during the parameter estimation (then not requiring uniform protocol across voxels, which is an important problem for high performance scanners such as MAGNUS and Connectome), the authors only compare their method with and without this correction. Why not compare against other methods from literature that also correct gradient non-uniformities? The authors cite one of them in reference 6 (same authors as the SMI method), its preprint version provide the code for the method, could the authors compare against this method and maybe other ones from literature.

Response

Thank you for the suggestion. The method was evaluated qualitatively by inspecting the regions where the correction had the strongest effect. The largest impact was observed at the anterior and posterior parts of the brain, which is consistent with expectations, as gradient non-uniformities are most pronounced in these regions. Nevertheless, we agree that a quantitative comparison with alternative methods would be valuable.

We investigated the method PIPE, as suggested by the reviewer and have updated the ISMRM abstract reference by the preprint citation. However, its implementation currently accommodates Linear Tensor Encoding (LTE), whereas the data presented in this manuscript also consists of non-LTE acquisitions. The authors mention rightfully that for non-LTE acquisitions, the symmetry of the diffusion tensor is not necessarily preserved. While we did correct for deviating b_{Δ} , we did not take into account differences that arise in the second and third eigenvalues of the B-tensor (b_{η}). We acknowledge that this is important to note, and we have now added this in the discussion section 4.2 (see lines 325 -327) *While changes in shape due to gradient non-uniformities are taken into account (b_{Δ}), a limitation of the current implementation is that it assumes conservation B-tensor axial symmetry, an assumption that generally does not hold when gradient non-uniformities and non-LTE encodings are considered (Coelho et al., 2025) The exact impact of this approximation would necessitate implementation of $SO(3)$ convolutions and requires further investigation.*

We considered performing a comparison with PIPE on the LTE subset of the data, but concluded that this would disadvantage the method due to the inherent degeneracies of the SM. To our knowledge, no other methods are currently available for gradient non-uniformity correction of SM parameters except NLLS.

To strengthen the evaluation, we included an additional experiment in which NLLS fitting was performed with a varying protocol per voxel. The results demonstrate that the effect is similar to that obtained with the INR-based gradient non-uniformity correction method (see Fig.R4). Furthermore, we have moved the section discussing the combination of Rician bias correction with gradient non-uniformity correction to the supplementary information section 5. This decision was motivated by the observation that there is little difference between the MSE and Rician bias correction approaches in this context, while the comparison with the NLLS method provides more relevant insight for the main text. We added the following text to the manuscript (see lines 238-242):

The effect of the gradient non-uniformity correction is similar for both INR and NLLS fitting. Lower diffusivity values at the front and back of the brain appear using both methods, as well as higher p_2 values. The parameter f shows only small differences between the approaches: NLLS shows no effect, while INR shows small corrections throughout the brain. Results of combining Rician bias loss with gradient non-uniformity correction can be found in the supplementary information.

R3.C8 Regarding the in vivo data in section 2.7, could you please describe in the manuscript the preprocessing steps? Preprocessing data with combined b, beta is not trivial, for example, did you use eddy? If so, how did you handle having same b-values with different betas, and the shells with very few directions (i.e. 3, 6, etc.)?

Response

We agree with the reviewer that this is not trivial to correct, e.g. with FSL eddy. We have used the multidimensional diffusion toolbox for correction, although the topup and gradient non-uniformity correction had to be performed separately. We have now added the description of the preprocessing to the methods (see lines: 166-167) *The in vivo data was corrected for Gibbs ringing (Kellner et al., n.d.), signal drift (Vos et al., 2017), motion and eddy current correction (Nilsson et al., 2015), susceptibility correction (Andersson et al., 2003), and gradient non-uniformity image distortion and B-matrix correction (Bammer et al., 2003)*

R3.C9 In Figure 6, could the ROI for the in vivo data match better the ROI for the synthetic data? At first impression it seems that for the in vivo data there are missing crossing fibers in comparison with the synthetic data, but at a closer look they seem to not be positioned at the same level, and I assume the intention of the authors was for them to match.

Response

We agree that it is preferable to have the images line up. The differences in slicing and especially the windowing (the in-vivo data does not contain the full brain, only the center slices) make this difficult. As a response on a comment by R1 (C7), we have added the SNR 20 synthetic dataset to Figure 6, and placed the in-vivo results in a separate figure (Figure 7).

R3.C10 In many cases, such as in the discussion and appendix B, authors mention method 1, method 2, method 3, never defined, instead of calling the methods by their name.

Response

We have updated the text to now include the method names instead of numbers.

R3.C11 While equation (1) is correct, why not writing it in integral form, as in most literature about Standard Model?

Response

We thank the reviewer for this suggestion. We implemented the suggestion accordingly.

R3.C12 Check for typos in the manuscript, just the abstract contains at least three typos, two in which there is no space between the period at the end of the sentence and the following letter, and another one in which besides the lack of the space there is not period at all, it says "existing methodsResults".

Response

The typos in the abstract have been fixed and we made sure to carefully proofread the manuscript to remove other typos.

5. Updated Figures

Figure R1: SM parameter maps fitted on in vivo data where each row corresponds to a different method.

Figure R2: Visualization of the centrum semi-ovale for different SH orders l_{max} (rows) and datasets (columns), with a parameter map of f_i as background. FODs are scaled for visibility.

Figure R3: Visualization of the centrum semi-ovale for different SH orders l_{max} of the in-vivo dataset, with a parameter map of f_i as background. FODs are scaled for visibility.

Figure R4: Effect of gradient non-uniformity correction on the estimation of SM parameters. Top row shows parameter maps without correction. Middle rows show the effect of gradient non-uniformity correction with INR. Bottom rows shows effect of gradient non-uniformity correction on NLLS parameter estimation. The difference maps are computed relative to the parameter estimates obtained with the same method, but without applying gradient non-uniformity correction.

Figure R5: Fitting methods comparison with in vivo protocol (SNR = 50). Scatter density plots of the ground truth versus the parameter estimations of all methods. Row numbers correspond to method numbering from section 2.5. Title of the subplots indicate ρ and RMSE. b) SM parameter maps corresponding to the results in a. Bottom row shows the ground truth (GT).

Figure R6: Scatter plots of ground truth versus SMI fits with different settings. Noise variance estimation: the standard approach used an MPPCA-derived sigma map, but since preprocessing introduces noise correlations, alternative estimates were examined, including b0-based standard deviation maps (1), MPPCA maps from b0 only (2), and omission of a sigma map (3). Acquisition selection: the impact of excluding $b = 200$ images (potential free-water contamination) and $b > 10,000$ images (low SNR) was evaluated (4). All SMI fits were executed with longer diffusion time ($\Delta = 49$ ms) to investigate a potential bias of choosing $\Delta = 19$ ms

Figure R7: Histograms of the SM parameter distributions using various settings and datasets, described in caption of Fig. R6.

Figure R8: Experiment 1 (SNR 50) without applying smoothing to the ground truth parameters: a) Scatter density plots of ground truth versus parameter estimations of all methods. The titles of the subplots indicate ρ and RMSE. b) SM parameter maps corresponding to the results in a. Bottom row shows the ground truth (GT).

References

- Andersson, J. L., Skare, S., & Ashburner, J. (2003). How to correct susceptibility distortions in spin-echo echo-planar images: application to diffusion tensor imaging. *NeuroImage*, *20*(2), 870–888. Retrieved from <https://www.sciencedirect.com/science/article/pii/S1053811903003367> doi: [https://doi.org/10.1016/S1053-8119\(03\)00336-7](https://doi.org/10.1016/S1053-8119(03)00336-7)
- Bammer, R., Markl, M., Barnett, A., Acar, B., Alley, M. T., Pelc, N. J., ... Moseley, M. E. (2003, 9). Analysis and generalized correction of the effect of spatial gradient field distortions in diffusion-weighted imaging. *Magnetic Resonance in Medicine*, *50*(3), 560–569. doi: [10.1002/mrm.10545](https://doi.org/10.1002/mrm.10545)
- Coelho, S., Lemberskiy, G., Zhu, A., Lee, H.-H., Abad, N., Foo, T. K. F., ... Novikov, D. S. (2025). *What if each voxel were measured with a different diffusion protocol?* Retrieved from <https://arxiv.org/abs/2506.22650>
- de Almeida Martins, J. P., Nilsson, M., Lampinen, B., Palombo, M., While, P. T., Westin, C. F., & Szczepankiewicz, F. (2021, 12). Neural networks for parameter estimation in microstructural MRI: Application to a diffusion-relaxation model of white matter. *NeuroImage*, *244*. doi: [10.1016/j.neuroimage.2021.118601](https://doi.org/10.1016/j.neuroimage.2021.118601)
- Dwedari, M. M., Consagra, W., Müller, P., Turgut, Ö., Rueckert, D., & Rathi, Y. (2024). Estimating neural orientation distribution fields on high resolution diffusion mri scans. In M. G. Linguraru et al. (Eds.), *Medical image computing and computer assisted intervention – miccai 2024* (pp. 307–317). Cham: Springer Nature Switzerland.
- Hendriks, T., Vilanova, A., & Chamberland, M. (2023). Neural spherical harmonics for structurally coherent continuous representation of diffusion mri signal. In *International workshop on computational diffusion mri* (pp. 1–12).

- Hendriks, T., Vilanova, A., & Chamberland, M. (2025). Implicit neural representation of multi-shell constrained spherical deconvolution for continuous modeling of diffusion mri. *Imaging Neuroscience*, *3*, imag_a__00501.
- Kellner, E., Dhital, B., Kiselev, V. G., & Reisert, M. (n.d.). Gibbs-ringing artifact removal based on local subvoxel-shifts. *Magnetic Resonance in Medicine*, *76*(5), 1574-1581. Retrieved from <https://onlinelibrary.wiley.com/doi/abs/10.1002/mrm.26054> doi: <https://doi.org/10.1002/mrm.26054>
- Nilsson, M., Szczepankiewicz, F., van Westen, D., & Hansson, O. (2015, 11). Extrapolation-based references improve motion and eddy-current correction of high b-value dwi data: Application in parkinson's disease dementia. *PLOS ONE*, *10*, 1-22. Retrieved from <https://doi.org/10.1371/journal.pone.0141825> doi: 10.1371/journal.pone.0141825
- Vos, S. B., Tax, C. M. W., Luijten, P. R., Ourselin, S., Leemans, A., & Froeling, M. (2017). The importance of correcting for signal drift in diffusion mri. *Magnetic Resonance in Medicine*, *77*(1), 285-299. Retrieved from <https://onlinelibrary.wiley.com/doi/abs/10.1002/mrm.26124> doi: <https://doi.org/10.1002/mrm.26124>
- Westin, C.-F., Szczepankiewicz, F., Pasternak, O., Ozarslan, E., Topgaard, D., Knutsson, H., & Nilsson, M. (2014). Measurement tensors in diffusion mri: generalizing the concept of diffusion encoding. In *Medical image computing and computer-assisted intervention – miccai 2014* (Vol. 8674, pp. 209–216). Springer. doi: 10.1007/978-3-319-10443-0_27
- Zhang, X., Duchemin, Q., Liu*, K., Gultekin, C., Flassbeck, S., Fernandez-Granda, C., & Assländer, J. (2022). Cramér-rao bound-informed training of neural networks for quantitative mri. *Magnetic Resonance in Medicine*, *88*(1), 436-448. Retrieved from <https://onlinelibrary.wiley.com/doi/abs/10.1002/mrm.29206> doi: <https://doi.org/10.1002/mrm.29206>
- Álvaro Planchuelo-Gómez, Descoteaux, M., Larochelle, H., Hutter, J., Jones, D. K., & Tax, C. M. (2024). Optimisation of quantitative brain diffusion-relaxation mri acquisition protocols with physics-informed machine learning. *Medical Image Analysis*, *94*, 103134. Retrieved from <https://www.sciencedirect.com/science/article/pii/S1361841524000598> doi: <https://doi.org/10.1016/j.media.2024.103134>

1. General comments to the reviewers

We thank the reviewers for yet another round of comments. The focus seems to have shifted to more manuscript structure related issues and some requests for elaboration. We have processed these comments and agree that it has further improved the structure and clarity of our manuscript. Revisions in the main manuscript are marked in red, while the changes made in the previous round remain blue.

2. Reviewer 1

In the revised version of the manuscript authors have provided answers to the questions raised in the previous review, and have modified passages of the text accordingly, and have added error computation results in the appendix section 7 (tables 1 and 2; formerly tables D4 and D5). It is unclear whether the results they have added as supplementary materials in its section 4 answer adequately to the questions raised. Also, the choice of the single low SNR value they have used to demonstrate their method is still not adequately justified.

R1.C1 Authors have split and renamed the supplementary material sections, which has made keeping track of changes a lot harder. Additionally, there is no color distinction between new content and previous content in the supplementary materials, including new supplementary material sections. Finally, they have used the same numbering scheme (e.g. without any prefix, using letters exclusively, etc.) for supplementary sections, figures and tables as those used in the main section. All this results in a complex task of checking the references within the rebuttal and the revised version of the manuscript. Should there be another round of reviews, please use a specific numbering for the supplementary material sections, figures and tables, and color new section titles, figure and table names and captions with the color chosen for the new text.

Response

We thank the reviewer for pointing this out. We acknowledge that the current organization of the supplementary material may make it more difficult to track changes, however we felt that splitting them from the main document (instead of appendices) was necessary with future publication in mind. We have ensured that the supplementary sections, figures, and tables are clearly distinguished from those in the main text by adopting a specific numbering scheme (prefixing with "S") and by using color highlighting for all newly added content and section titles, as suggested.

R1.C2 Related to R1.C1, authors have still not shown how the protocol generalizes to sparsely sampled datasets, and still keep the argument that motivated the question ("INRs demonstrate potential to improve (...) in sparsely sampled datasets."). They explicitly say "Although not a further reduced protocol, these experiments show generalizability to other acquisition protocols". So, reading the description of section 4 of the appendix related to this experiment, it becomes apparent that the acquisition protocol (table 2) is still not what we would call sparse. To be clear, it is informative for the community/readership the sampling scheme where your method results starts to degrade. Similarly, authors chose SNR 50 to demonstrate results, which again may not be considered a low SNR setting. In that response authors say that studying sparsely acquired datasets is out of the scope of the work: if such is the case, the passage mentioning the potential of INRs to improve estimation on sparse acquisitions should be removed and be discussed.

Response

We thank the reviewer for their comment. The sentence in question refers to previous work (Hendriks et al., 2023), which does explore ablation of the protocol (albeit for a different dMRI model). To clear up confusion and make the sentence more applicable to this specific work we adapted the sentence to only specify noise. While we agree with the reviewer that designing sparse protocols is interesting, fitting of the Standard Model, even on high-quality acquisitions, is a non-trivial task as evidenced by several previous works and the results in this work, and methods are still being developed. We can see how the performance for other methods already starts to degrade at the SNR=50 level. Once solid methods have been established for high-quality data, one can start exploring further ablation of protocols. We have removed all claims of applying to sparsely sampled data and added a passage to the discussion stating that this should be explored. Line number 403: "The performance of INRs for sparse, clinically feasible acquisition protocols remains to be investigated and represents a direction for future research (Álvaro Planchuelo-Gómez et al., 2024)"

R1.C3 Related to R1.C3, the SNR 25-100 range that authors employ to justify that SNR 20 and 50 are representative of noisy settings is still not convincing enough or not backed. Can authors provide references to works that clearly state that the SNR below 25 in DWI is extremely rare?

Response

Various works investigate the SNR in dMRI. For high-resolution (and therefore, usually lower SNR) the $b = 0$ SNR rarely drops below 20 (Choi et al., 2011; Feizollah & Tardif, 2023). In other papers, more closely related to this work, authors mention (without reference) "clinically relevant SNR values of 20-30" (Tournier et al., 2013) and do their experiments on a minimum of SNR = 20. The related paper on SMI only drops the SNR to 40 for their experiments, see appendix of (Coelho et al., 2022), and the supervised method by Almeida Martins simulates noise in the SNR 80-120 range on the $b = 0$ (de Almeida Martins et al., 2021). Also of interest regarding this matter might be Ref. (Liao et al., 2024) which investigates SM fitting in clinical setting and uses SNR 20 and 25 for simulations. We would kindly like to re-iterate the statement in R1.C2 that our goal is not to find the edges at which fitting the Standard Model is still possible, but rather to fit the model in scenarios (i.e. optimized protocols and noise settings) similar to those achieved in previous work and associated realistic acquisitions. We can see that in SNR=50 most established methods already start losing performance, and at SNR=20 significantly degrade and in some cases become unusable (e.g. NNLS). Future work could explore to lower the SNR further, but to maintain focus we consider this beyond this paper. We have added a line clarifying this in the methods. Line 174: "These SNR levels are comparable to (50), or below (20), those investigated in previous work (Coelho et al., 2022; de Almeida Martins et al., 2021)."

R1.C4 Related to R1.C7, reviewers cannot know what Figure R2 and Figure R3 refer to, as these names do not exist in the revised version of the manuscript.

Response

Apologies for not stating this clearly. The figures names starting with 'R' were the figures in the response to reviewer document, which where put there conform guidelines, and for the reviewers convenience. They correspond to the figures mentioned in the reviewers comment (6 and 7 in the current manuscript).

R1.C5 Related to R1.C8, similar to the previous comment and comment 1, tables D.4 and D.5 are no longer present in the revised manuscript. In this case, given that they are the

only tables in the supplementary materials, the reviewer has been able to locate them, but future rebuttals and revised versions should avoid these issues.

Response

Again our apologies for this inconvenience.

R1.C6 Related to R1.C9, the question asked referred to the inference time, not the fitting time: authors have pointed to Table 3, which is about fitting. Thus, the question remains to be answered.

Response

Thank you for this clarification, and sorry for the misunderstanding. Since only the last layer of the model changes the inference times are not dramatically different. We have updated the sentence (line 305) to include the time for $l_{max} = 8$, which is 2.8 seconds for 1 million inferences. This evaluation is added to the discussion.

R1.C7 Related to C1.C12, then the two appearances of the term "B-tensor" in line 58 of the revised manuscript should also be capitalized.

Response

This is indeed the case, we have fixed these typos.

R1.C8 Related to R1.C16, even if the worded description in the rebuttal is clear, the notation is still not and the passage in the manuscript is still misleading. Please, make the necessary changes so that the notation and meaning is not misleading.

Response

We appreciate the reviewer's observation. We have clarified the notation to avoid ambiguity and ensure consistency with the text. Specifically, the extra-axonal diffusivities are now denoted as D_e^{\parallel} (parallel) and D_e^{\perp} (perpendicular) throughout the manuscript.

R1.C9 Related to R1.C17, the equations are still not right: for Eq 2: please check Tax et al. Neuroimage 236 (2021), equation A.3: the brackets are either not OK, or the b multiplier in the right-most operand is now redundant; for Eq 3: the exponent of the term exp

$$(1/3bD_i - 1/3bD_i)$$

is now zero, which was not the case previously, and b_{delta} has disappeared from the right-most term. Please fix this or explain if there was an error in the previous version of the manuscript.

Response

Thank you for pointing this out. Regarding Eq 2: indeed a bracket was missing for the term $D_{\parallel} + 2D_{\perp}$ while there was a redundant bracket at the end of the expression. This is now fixed. Regarding Eq 3: b_{delta} is reinserted, also fixing the fact that the specific term would be zero.

R1.C10 Related to R1.C27, a form of the response should be added to the discussion.

Response

We agree that this is an interesting point to discuss and have added it to the discussion section of our manuscript. Lines 346-350: "The INR shows a slight overestimation in D_e^{\parallel} in the splenium of the corpus callosum in the synthetic experiments. This could be due to the ground truth exhibiting less structural coherence in this part, which is especially apparent in the D_e^{\perp} parameter map. Since the parameters are estimated jointly this might influence the estimation of D_e^{\parallel} . Potentially, SMI does not suffer from this because it fits the SM voxel-wise and is, therefore, able to produce these combinations of parameters."

R1.C11 Related to R1.C28, authors answer that "The INR could get closer to the ground truth in noiseless scenarios by increasing sigma and the MLP size.". Can this be demonstrated with an experiment? Otherwise, the argument is void.

Response

The results in the supplementary material 1 point in this direction, although they are not quantified. We have removed the statement about sigma as this is not demonstrated as you correctly pointed out.

R1.C12 Related to R1.C29, in response to the question, authors say that they have changed the figure caption. Unfortunately, it is not the case. Please, change the caption to avoid confusion.

Response

We have revised the figure regarding the gradient non-uniformity correction to include NLLS and clarified the description regarding the difference maps per your suggestion (see blue text caption Fig.8). The original figure was moved to the supplementary. We have adopted the suggested caption change (see Fig.S6)

R1.C13 In response to comment C30 (R1.C30), by removing the referenced to the figure, authors have left a statement that is not supported by any numerical or graphical result, as they only show WM data, which is not helpful. Please, provide references or elaborate on the statement so that readers that have not run any INR model can understand exactly the limitation that they are referring to.

Response

We have expanded the explanation in the discussion to be more clear. Line 383-387: "(...) fits the Standard Model of white matter, which – as the name suggests – is only applicable for white matter. As a result, we applied a white matter mask, and only used the coordinates that lie inside this mask as input to the INR. This implies that for coordinates outside of the mask, and in different tissue types, INR results are not fit (correctly)."

R1.C14 Line 149: there is a typo in the passage "(...) and 1 Nlevels."

Response

Thank you for pointing this out, we have fixed the formatting to be more clear.

R1.C15 Line 213: "Parameter estimation on simulated data without noise and with SNR = 50 can be found in the supplementary information (...)." Please, point to the specific section/figure.

Response

We have included the section number in the text.

R1.C16 Line 235: "This holds true for all kernel parameter estimates, which is shown in further detail in the supplementary information section 6." Did authors mean section 7 instead of section 6? Related to major comment 1, please, add a prefix (e.g. S) or use letters exclusively to number the supplementary material sections so that it becomes unequivocal when authors are referring to them. Same goes to the figures and tables.

Response

We indeed meant section 7. Again our apologies for the unclear structure. We have made sure to improve it.

R1.C17 It looks like section 4/figure 5 of supplementary materials is not cross-referenced in the manuscript. Or it is not evident, at least.

Response

We have added a reference to these results in Results section 3.3. Thank you for pointing this out.

R1.C18 The section 3.6 text should appear before the figure that it explains (Figure 9).

Response

Thank you, we have checked that this is the case.

R1.C19 Line 269: "Parameter estimates for FOD up to SH order eight can be provided alongside (...)." The sentence suggests that parameters of larger SH orders cannot be provided. Although in their experiments authors employ SH orders up to 8, this should not mean that higher order coefficients cannot be provided. Please, reword the passage.

Response

We have adapted the text to state 'at least eight' instead of 'up to eight'.

R1.C20 Line 284: l_{\max} : Please, use the same notation used throughout the rest of the manuscript, making " \max " be a subscript.

Response

Thank you for pointing this out. We noticed the discrepancy after resubmitting and have fixed it.

R1.C21 Line 305: the sentence "The lower performance of the INR method on simulated data without noise arises from its inability to capture the input data perfectly, as it does not model individual voxels, unlike the other methods." is misplaced as it is written between two sentences that are related to each other. Please, move the sentence to a better location in the passage.

Response

Thank you for the suggestion. We believe the sentence is correctly placed because it comments directly on the performance of the INR method introduced in the preceding sentence. To improve clarity and flow we have rewritten the sentence and left it in its current position. See line 325: "The INR method exhibits lower performance on noiseless signals due to its inability to model voxels individually, indicating that the ground truth is not positively biased with respect to the outputs of the INR method."

R1.C22 Line 311: "We have investigated the impact of other possible sources of severe bias on creating ground truth parameter maps from the MGH dataset, such as the relatively short diffusion time, noise estimation procedure, and included b-values (results not shown)". Please show those results as supplementary materials.

Response

We have added the information in supplementary information section 8. And have modified the discussion accordingly (see line 333).

R1.C23 Line 315: please, fix the typo in " (\dots) and $b_{\perp} 10.000 \text{ s/mm}^2$ ".

Response

We have changed the mentions of b-value to be in inline equations, which hopefully solves the issue.

R1.C24 Line 333: the sentence ending period is missing in "(...) encodings are considered (7) The exact (...)".

Response

Thank you for your attention to detail. We have fixed this typo.

3. Reviewer 2

This paper presents a deep learning-based method for signal estimation, grounded in the Standard Model of white matter. Their work is inspired by the effectiveness of implicit neural representations (INR)—a more general deep learning technique for representing a signal, e.g. an image, as a continuous function (i.e. rather than discrete pixels). They evaluate their method across both synthetic and in-vivo datasets. They compare their approach to 3 other methods and find their INR approach delivers broadly superior results.

I would like to thank the authors for their extensive work to reply to and make corrections based on the feedback from all reviewers. Below I include a response to the changes made as well as additional comments on other items to improve. I believe the authors have largely addressed my points, with some caveats on comment 2. Beyond this, I have included additional comments that are largely areas in the paper that would benefit from additional clarification, as well as a few minor typographical issues.

R2.C1 I thank the authors for updating the usage of unsupervised to self-supervised. I believe this is now more in-line with standardised usage, while still emphasising the novelty of the approach compared to previous supervised techniques.

Response

We agree that it's an improvement and will use self-supervised in future research.

R2.C2 I thank the authors also for their discussion around the effects of SNR and how INR appears to show greater robustness in lower SNR regimes. However, as I see it, there are two core innovations in this paper: the application of a continuous, rather than discrete representation over voxels (INR) and the self-supervised approach that discovers the SM parameters. From here, it is not obvious which is providing the core benefit here. If we imagine self-supervised INR as one quadrant, then supervised INR and voxel-based self-supervised would be other possible ways to approach the problem. The authors indicate in their reply

that a supervised INR would be impractical due to the one model per-dataset fitting. However, even if both these quadrants are impractical it would be useful to clarify this in the paper, as otherwise it isn't clear to me at least why the baseline supervised NN from (13), without being deeply familiar with that paper, couldn't also use INR.

Response

Thank you for this interesting point of discussion. INRs are inherently self-supervised, and make use of spatial correlations that are specifically present in one particular dataset. Now, there exists a counterpart of the INR called the Explicit Neural Representation (ENR). This model learns, as you suggest, a continuous representation by using a training set in a supervised way. To make it applicable to other datasets the ENRs are conditioned during training using the training set data. Then, conditioning the INR using an unseen example, one can generate a new continuous output. An example of this in diffusion MRI is called FENRI Spears & Fletcher (2024). In our opinion self-supervised learning provides benefits because it is not imposing training set biases on to the unseen data, and the continuous, spatially regularized representation provides benefits in parameter estimation and coherence between voxels. We have added a sentence clarifying this in the discussion (see line 281-283): "The self-supervised, subject-wise nature of the framework prevents training set bias, while the continuous representation allows spatial correlations to improve parameter estimates and reduce the impact of noise."

R2.C3 Based on R1 C6, I think it would be useful to clarify in the paper the lack of train / test split in the paper. As far as I can see, INRs do typically have a train / test split (e.g. Mildenhall et al., 2020. NeRF: Representing Scenes as Neural Radiance Fields for View Synthesis). I understand why the split may not matter as much in this case due to the lack of direct supervision on the SM parameters, but may be worth clarifying in the paper.

Response

Thank you for this question. The train / test in Mildenhall is due to novel view synthesis being their goal. It would be akin to generating DWI volumes in unseen b-vector directions in our work, or possibly upsampling spatially where the target is known. Since we do neither of these experiments we have specifically steered away from using train/test terminology, as not to confuse readers with a supervised approach. We have added a sentence to clarify this in the methods section 2.3 (see line 125-126): "The full dMRI dataset is used, without a train/test split, as the goal is for the INR to represent the data, not to predict unseen data."

R2.C4 Related to this, did the authors observe any impact to the SM parameters from over / underfitting to the dataset? Is there a way to evaluate the SM parameters mid-training

run to implement early stopping?

Response

Thank you for another valid and interesting question. We have looked at this in initial experiments with randomly excluding spatial coordinates from the fitting process. The model performance on these unseen voxels could be used as a 'validation' score during fitting and used for early stopping. Overfitting (where the validation loss started increasing) was not observed in the cases investigated. We chose our number of epochs based on when the training loss converges, which more-or-less lined up with when the validation loss converged.

R2.C5 Line 25: I think it would help to be specific what is being supervised or not in the case of self-supervised, and how the self-supervised case only relies on the signal.

Response

We have added an explanation about the supervised and self-supervised training data in the introduction. See line 28-29 and line 33-34

R2.C6 Line 32: INR is described as different from voxel-based methods, but it isn't clear to anyone unfamiliar with INR why it is different. This is further made confusing by the reference to 'per voxel' in line 46.

Response

We have adapted the sentence to be more clear that INRs operate in a continuous coordinate space, and changed per voxel to per coordinate.

R2.C7 There is some inconsistency around use of dashes throughout the paper, e.g. in-vivo versus in vivo, noise-robustness versus noise robustness.

Response

Thank you for pointing this out. We searched or document for '-' and a couple more inconsistencies came up, which we have fixed together with your two suggestions.

R2.C8 Line 49: It is unclear why the proposed approach is not biased by the training data. I believe this is discussed slightly more later in the paper, but is omitted here.

Response

We have made the clarification that the training is self-supervised and therefore not biased by training data.

R2.C9 Line 57: Should there be punctuation after 'on the unit sphere'?

Response

We have checked the punctuation in from of equations and made it consistent throughout the manuscript.

R2.C10 Line 98: Sigmoid shouldn't be capitalised here as far as I'm aware.

Response

We have changed it to be a lower-case 's'.

R2.C11 Line 143/144: Appears to be a strange line break in the notation here.

Response

Fixed.

R2.C12 I think there needs to be more clarity on the usage of 'dataset' through the paper. As I understand for INR, one subject = one dataset, but this may be confusing to readers given a dataset often refers to a set of subjects for many machine learning tasks.

Response

Thank you for pointing this out. We tried to find unambiguous wording. We have gone over the manuscript again, searching for the word 'dataset', and cleaned up any ambiguous formulations. Additionally we added a sentence in the 'Generation of simulated ground truth data' paragraph, that clarifies what we mean by the word 'dataset'.

R2.C13 In the future work section, it could be worth including the possibility of extending the work to a multi-dataset (i.e. multi-subject) setup, where the model parameters are shared across subjects (see Wang et al., 2024. SCARF: Scalable Continual Learning Framework for Memory-efficient Multiple Neural Radiance Fields, for instance). This seems to be a natural next step to get sublinear increases in train time and model size as the number of subjects increases.

Response

We thank the reviewer for this interesting perspective, and we have added meta-learning and continual learning as a possibility to decrease fitting time to the discussion with the following citation (Tancik et al., 2021; Wang et al., 2024), alongside the other strategies that we already mentioned in the previous manuscript iterations.

4. Reviewer 3

I thank the authors for addressing my comments satisfactorily, in particular, their comprehensive evaluation of the Ground Truth generation (removing the smoothing step and exploring different

sigma maps and protocols), and although they did not perform the comparison I suggested, I am satisfied with the comparison against NLLS fitting with varying protocol per voxel and the discussion about the non-trivial limitation of their gradient non-uniformity correction approach.

I do not have further comments

Response

We thank the reviewer for their positive assessment.

References

- Choi, S., Cunningham, D. T., Aguila, F., Corrigan, J. D., Bogner, J., Mysiw, W. J., ... Schmalbrock, P. (2011). Dti at 7 and 3 t: systematic comparison of snr and its influence on quantitative metrics. *Magnetic resonance imaging*, 29(6), 739–751.
- Coelho, S., Baete, S. H., Lemberskiy, G., Ades-Aron, B., Barrol, G., Veraart, J., ... Fieremans, E. (2022, 8). Reproducibility of the Standard Model of diffusion in white matter on clinical MRI systems. *NeuroImage*, 257. doi: 10.1016/j.neuroimage.2022.119290
- de Almeida Martins, J. P., Nilsson, M., Lampinen, B., Palombo, M., While, P. T., Westin, C. F., & Szczepankiewicz, F. (2021, 12). Neural networks for parameter estimation in microstructural MRI: Application to a diffusion-relaxation model of white matter. *NeuroImage*, 244. doi: 10.1016/j.neuroimage.2021.118601
- Feizollah, S., & Tardif, C. L. (2023). High-resolution diffusion-weighted imaging at 7 tesla: Single-shot readout trajectories and their impact on signal-to-noise ratio, spatial resolution and accuracy. *Neuroimage*, 274, 120159.
- Hendriks, T., Vilanova, A., & Chamberland, M. (2023). Neural spherical harmonics for structurally coherent continuous representation of diffusion mri signal. In *International workshop on computational diffusion mri* (pp. 1–12).
- Liao, Y., Coelho, S., Chen, J., Ades-Aron, B., Pang, M., Stepanov, V., ... Fieremans, E. (2024, 3). Mapping tissue microstructure of brain white matter in vivo in health and disease using diffusion MRI. *Imaging Neuroscience*, 2, 1–17. doi: 10.1162/imag{_}_a{_}_00102
- Spears, T., & Fletcher, P. T. (2024). Learning spatially-continuous fiber orientation functions. In *2024 ieee international symposium on biomedical imaging (isbi)* (pp. 1–5).
- Tancik, M., Mildenhall, B., Wang, T., Schmidt, D., Srinivasan, P. P., Barron, J. T., & Ng, R. (2021). Learned initializations for optimizing coordinate-based neural representations.

In *Proceedings of the IEEE/CVF Conference on Computer Vision and Pattern Recognition* (pp. 2846–2855).

Tournier, J.-D., Calamante, F., & Connelly, A. (2013). Determination of the appropriate b value and number of gradient directions for high-angular-resolution diffusion-weighted imaging. *NMR in Biomedicine*, 26(12), 1775–1786.

Wang, Y., Wang, J., Wang, C., Duan, W., Bao, Y., & Qi, Y. (2024). Scarf: Scalable continual learning framework for memory-efficient multiple neural radiance fields. In *Computer graphics forum* (Vol. 43, p. e15255).

1. General comments to the reviewers

Thank you for your incredibly detailed review of this manuscript. Review processes like these improve the quality of the reviewed work and help maintain a high standard for scientific publications. The final comment correctly points out a difference in equations between a published manuscript and this one, where the equation in this work is actually the correct one. The author of the published manuscript is in the process of rectifying the mistake in their paper. No further changes to our manuscript have been made.

2. Reviewer 1

Authors have appropriately addressed the questions raised in the previous review. I thank them for their effort and work. I am only pointing to an error that persists across versions in equations 2 and 3.

R1.C1 In Eq.2, the parallel and perpendicular components of the second operand are swapped (they have been swapped since the first version of the manuscript) when comparing to Tax et al. 2021, where it is written as $1/3 * b * (D_{perpendicular} + 2 * D_{parallel})$. Is there a reason for the swap? If this is an oversight, when fixing it, please note that Eq. 3 will also need to be changed accordingly.

Response

The equation in Tax et al. 2021 is wrong and the author is in the process of rectifying this mistake. The equation in this manuscript is the correct one. We have, therefore, made no further changes. Thank you for your attention to detail (throughout this whole review process)!

3. Reviewer 2

This paper presents a deep learning-based method for signal estimation, grounded in the Standard Model of white matter. Their work is inspired by the effectiveness of implicit neural representations (INR)—a more general deep learning technique for representing a signal, e.g. an image, as a continuous function (i.e. rather than discrete pixels). They evaluate their method across both synthetic and in-vivo datasets. They compare their approach to 3 other methods and find their INR approach delivers broadly superior results.

I would like to thank the authors for the additional changes and feedback following the second round of reviews. I believe all of my concerns raised and clarifications requested have

now adequately addressed by the authors.

I have no further comments.

Response

We thank the reviewer for the time and effort that went into reviewing this paper.

References